# TGF-β uses a novel mode of receptor activation to phosphorylate SMAD1/5 and induce epithelial-to-mesenchymal transition

Anassuya Ramachandran[1], Pedro Vizán[1†], Debipriya Das[1‡], Probir Chakravarty[2], Janis Vogt[3], Katherine W Rogers[4], Patrick Müller[4], Andrew P Hinck[5], Gopal P Sapkota[3], Caroline S Hill[1]*

[1]Developmental Signalling Laboratory, The Francis Crick Institute, London, United Kingdom; [2]Bioinformatics and Biostatistics Facility, The Francis Crick Institute, London, United Kingdom; [3]Medical Research Council Protein Phosphorylation and Ubiquitylation Unit, University of Dundee, Dundee, United Kingdom; [4]Friedrich Miescher Laboratory of the Max Planck Society, Tübingen, Germany; [5]Department of Structural Biology, University of Pittsburgh School of Medicine, Pittsburgh, United States

**\*For correspondence:**
caroline.hill@crick.ac.uk

**Present address:** †Center for Genomic Regulation, Barcelona, Spain; ‡Flow Cytometry, The Francis Crick Institute, London, United Kingdom

**Competing interests:** The authors declare that no competing interests exist.

**Abstract** The best characterized signaling pathway downstream of transforming growth factor β (TGF-β) is through SMAD2 and SMAD3. However, TGF-β also induces phosphorylation of SMAD1 and SMAD5, but the mechanism of this phosphorylation and its functional relevance is not known. Here, we show that TGF-β-induced SMAD1/5 phosphorylation requires members of two classes of type I receptor, TGFBR1 and ACVR1, and establish a new paradigm for receptor activation where TGFBR1 phosphorylates and activates ACVR1, which phosphorylates SMAD1/5. We demonstrate the biological significance of this pathway by showing that approximately a quarter of the TGF-β-induced transcriptome depends on SMAD1/5 signaling, with major early transcriptional targets being the *ID* genes. Finally, we show that TGF-β-induced epithelial-to-mesenchymal transition requires signaling via both the SMAD3 and SMAD1/5 pathways, with SMAD1/5 signaling being essential to induce ID1. Therefore, combinatorial signaling via both SMAD pathways is essential for the full TGF-β-induced transcriptional program and physiological responses.
DOI: https://doi.org/10.7554/eLife.31756.001

## Introduction

Members of the transforming growth factor β (TGF-β) family of ligands, which includes the TGF-βs, Activins, NODAL, BMPs and GDFs, have pleiotropic effects on cell behavior ranging from germ layer specification and patterning in embryonic development, to tissue homeostasis and regeneration in adults (*Massagué, 2012*; *Morikawa et al., 2016*; *Wu and Hill, 2009*). TGF-β family signaling is also deregulated in a number of human diseases through mutation or altered expression of either the ligands or downstream signaling pathway components (*Miller and Hill, 2016*). In this context, the most widely studied pathology is cancer (*Bellomo et al., 2016*; *Massagué, 2008*; *Meulmeester and ten Dijke, 2011*; *Wakefield and Hill, 2013*), where TGF-β itself has both tumor suppressive and tumor promoting effects (*Massagué, 2008*). At early stages of cancer TGF-β's tumor suppressive effects dominate, such as its cytostatic and pro-apoptotic functions (*Padua and Massagué, 2009*). As tumors develop, however, mutations in key components of the pathway or downstream target genes allow the tumor to evade TGF-β's tumor suppressive effects, whilst remaining sensitive to its

**eLife digest** Cells communicate with other cells via signaling molecules to coordinate their activities. Signals released from one cell can influence the behavior of neighboring cells. Signaling molecules belonging to the TGF-β family play crucial roles in animals. For example, these molecules guide the formation of tissues and organs and help maintain them throughout the animal's adult life. Abnormal regulation of TGF-β family signaling can fuel the growth of cancer cells and also contribute to other diseases in humans.

Molecules in the TGF-β family bind to and bring together specific receptors on the surface of the receiving cell. This allows the receptors to activate so-called SMAD proteins within that cell. Activated SMADs move to the cell's nucleus, where they regulate the activity of target genes. This in turn changes how the cell behaves.

The best-studied member of the TGF-β family is TGF-β itself. It is well known to activate two particular SMAD proteins called SMAD2 and SMAD3. Recent research showed that TGF-β could also activate two different SMAD proteins, SMAD1 and SMAD5. However, it was not understood how this was achieved, or what its biological consequences were.

Ramachandran et al. set out to address these questions in mouse and human cells grown in the laboratory. The experiments showed that, in addition to its known dedicated receptors, TGF-β also requires a third receptor to activate SMAD1 and SMAD5. Also, TGF-β signaling leads to changes in the activity of several thousand genes, and approximately a quarter of them require signaling via SMAD1 and SMAD5.

Further work showed that SMAD1 and SMAD5 are needed for a process called epithelial-to-mesenchymal transition. This is a normal part of animal development, and is also a common feature of cancer cells, allowing them to spread to distant parts of the body.

Understanding of how TGF-β signaling works in more detail may reveal new ways to target this pathway to treat diseases like cancer. The next step is to see how the signaling via SMAD1 and SMAD5 contributes to different aspects of cancer development.

DOI: https://doi.org/10.7554/eLife.31756.002

tumor-promoting activities. TGF-β directly promotes the oncogenic potential of tumor cells, for example by driving epithelial-to-mesenchymal transition (EMT), a hallmark of cancer that enhances cell invasion and migration, and also increases the frequency of tumor-initating cancer stem cells (*Massagué, 2008*; *Ye and Weinberg, 2015*). TGF-β's dual role in cancer thus provides an excellent example of how diverse responses can be elicited by a single ligand.

The TGF-β family ligands all signal via a common mechanism, initiated by ligand binding to two cell surface serine/threonine kinase receptors, the type II and type I receptors. In the receptor complex, the type II receptors phosphorylate and activate the type I receptors (*Wrana et al., 1994*). These in turn phosphorylate the downstream effectors of the pathway, the receptor-regulated SMADs (R-SMADs) on two serines in an SXS motif at their extreme C-termini. Phosphorylated R-SMADs form complexes with the common SMAD, SMAD4, which accumulate in the nucleus and directly regulate the transcription of target genes, leading to new programs of gene expression (*Shi and Massagué, 2003*). In the classic view of TGF-β family signaling, there are two branches, characterized by distinct combinations of type II and type I receptors, and the recruitment of specific R-SMADs to particular type I receptors (*Wakefield and Hill, 2013*; *Shi and Massagué, 2003*). One branch is activated by TGF-β, Activins and NODAL and is mediated via the type I receptors TGFBR1, ACVR1B and ACVR1C (also known as ALK5, ALK4 and ALK7 respectively), which phosphorylate SMAD2 and 3. The other is activated by BMPs and GDFs and is mediated via ACVRL1, ACVR1, BMPR1A and BMPR1B (also known as ALK1, ALK2, ALK3 and ALK6 respectively), which phosphorylate SMAD1, 5 and 9 (*Miller and Hill, 2016*).

In general, while this pairing between type I receptors and R-SMADs broadly fits the assignment of specific ligands to the different branches of TGF-β family signaling, it is an oversimplification. For example, ACVR1 is now described as a BMP receptor, but early work indicated that it could bind Activin and TGF-β (*Massagué, 1996*; *Miettinen et al., 1994*), and very recently it has been shown to signal downstream of Activin in the context of the disease, fibrodysplasia ossificans progressiva

(*Hatsell et al., 2015*; *Hino et al., 2015*). Furthermore, ACVRL1, a type I receptor that recognizes BMP9 and 10, also transduces TGF-β signals in endothelial cells (*Pardali et al., 2010*) by phosphory-lating SMAD1/5 in parallel with the canonical TGF-β-induced phosphorylation of SMAD2/3 (*Goumans et al., 2002*; *Goumans et al., 2003*). This SMAD1/5 arm of TGF-β signaling has also been shown to occur in a wide range of other cell types, including epithelial cells, fibroblasts and cancer cell lines, which do not express ACVRL1 (*Liu et al., 1998*; *Daly et al., 2008*; *Liu et al., 2009*; *Wrighton et al., 2009*).

Important questions concerning this noncanonical TGF-β-induced SMAD1/5 phosphorylation remain unanswered. First, the mechanism by which TGF-β induces SMAD1/5 phosphorylation, in par-ticular, the type I receptors involved, is not known. Some studies have concluded that the canonical TGF-β receptors TGFBR1 and TGFBR2 are sufficient for phosphorylation of both SMAD2/3 and SMAD1/5 (*Liu et al., 2009*; *Wrighton et al., 2009*). In contrast, others demonstrated that one of the classic BMP type I receptors (ACVR1 or BMPR1A), or in endothelial cells, ACVRL1, is additionally required (*Daly et al., 2008*; *Goumans et al., 2002*; *Goumans et al., 2003*). The second crucial issue concerns the biological relevance of TGF-β-induced SMAD1/5 signaling. Nothing is known about the transcriptional program activated by this arm of TGF-β signaling, or indeed, the specific SMAD com-plexes involved. It is also not known to what extent this pathway is required for the physiological responses to TGF-β.

Here, we dissect the SMAD1/5 arm of TGF-β signaling and define the underlying mechanism and its biological function. We show that TGF-β-induced SMAD1/5 phosphorylation requires both TGFBR1 and ACVR1 and using biosensors, and an optogenetic approach, we establish a new para-digm for TGF-β receptor activation. We have mapped the binding sites on chromatin of nuclear phosphorylated SMAD1/5 (pSMAD1/5) genome-wide, which led us to define the target genes regu-lated by this arm of TGF-β signaling. We go on to show that this arm of signaling is required for TGF-β-induced EMT. Our data reveal that the full transcriptional programme activated in response to TGF-β requires integrated combinatorial signaling via both the SMAD2/3 and SMAD1/5 pathways.

## Results

### The kinetics of TGF-β-mediated SMAD1/5 phosphorylation

To begin to dissect which receptors are required for TGF-β-induced SMAD1/5 phosphorylation, we compared the kinetics of SMAD1/5 and SMAD2 phosphorylation in response to TGF-β. Using the human breast cancer cell line MDA-MB-231 and the mouse mammary epithelial cell line NMuMG as model systems, we found that TGF-β induced only transient phosphorylation of SMAD1/5 that peaked at 1 hr and then returned to baseline (*Figure 1A*). This was in contrast to a more sustained TGF-β-induced SMAD2 phosphorylation, or SMAD1/5 phosphorylation in response to BMP4. How-ever, transient SMAD1/5 phosphorylation is not a defining characteristic of this arm of TGF-β signal-ing, as another human breast cancer line, BT-549, exhibited sustained SMAD1/5 phosphorylation that is still readily detectable 8 hr after TGF-β stimulation (*Figure 1—figure supplement 1A*). Fur-thermore, when BT-549 cells were grown as non-adherent spheres, TGF-β-induced SMAD1/5 phos-phorylation did not attenuate at all in the first 8 hr of signaling (*Figure 1—figure supplement 1A*). Thus, the kinetics of TGF-β-induced SMAD1/5 phosphorylation are cell-type-specific and dependent on the culture conditions and are independent of the kinetics of TGF-β-induced SMAD2/3 phosphor-ylation, suggesting a distinct receptor complex may be involved.

To address whether new protein synthesis was required for the transient nature of TGF-β-induced SMAD1/5 phosphorylation, cells were induced with TGF-β in the presence of either a translation inhibitor, cycloheximide or a transcription inhibitor, actinomycin D. Inhibition of translation was unin-formative because it also led to a very rapid loss of TGFBR2 and TGFBR1, due to their short half-lives (*Vizán et al., 2013*). Use of actinomycin D, however, circumvented this problem, as *TGFBR2* and *TGFBR1* mRNAs are relatively stable (*Figure 1—figure supplement 1B*) and their translation was unimpeded. In these conditions, SMAD1/5 phosphorylation was sustained (*Figure 1B*; *Figure 1—fig-ure supplement 1C*). Thus, the rapid loss of pSMAD1/5 at later time points after TGF-β stimulation requires new transcription, suggesting that it is mediated by a component whose expression is induced by TGF-β.

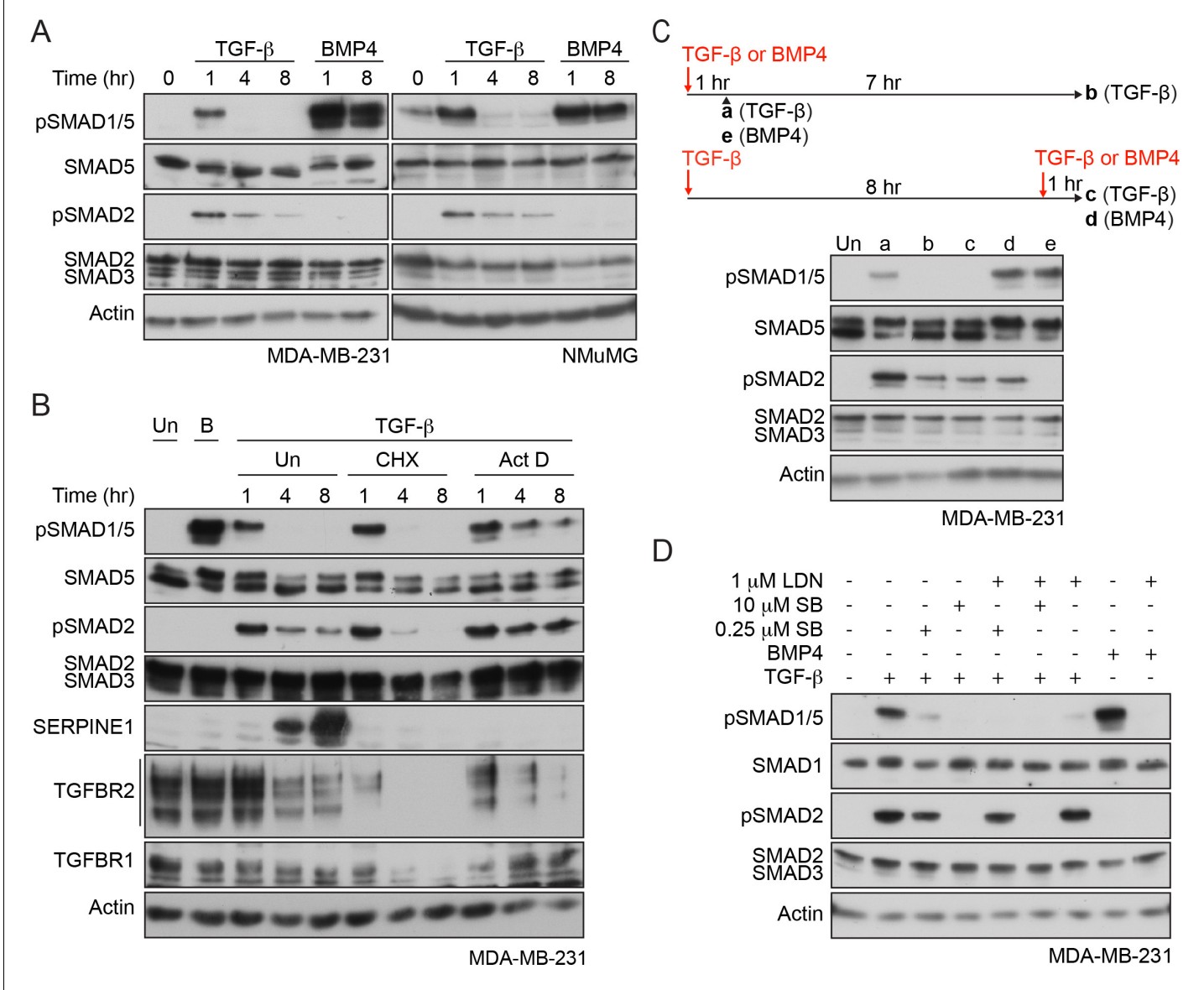

**Figure 1.** Characterization of SMAD1/5 phosphorylation by TGF-β. (**A**) MDA-MB-231 and NMuMG cells were treated with TGF-β or BMP4 for the times indicated. (**B**) MDA-MB-231 cells were treated with TGF-β for the times shown either alone or after 5 min pre-treatment with cyclohexamide (CHX) or actinomycin D (Act D). (**C**) MDA-MB-231 cells were treated with TGF-β for 1 or 8 hr, and after 8 hr, cells were re-stimulated with TGF-β or BMP4 for 1 hr as shown in the scheme. For comparison, cells were stimulated for 1 hr with BMP4. (**D**) MDA-MB-231 cells were induced or not with TGF-β or BMP4 in the presence of either 0.25 μM or 10 μM SB-431542 (SB) or 1 μM LDN-193189 (LDN) or a combination of 0.25 μM or 10 μM SB-431542 and 1 μM LDN-193189. In all panels, western blots are shown probed with the antibodies indicated. B, BMP4, Un, unstimulated. In B, SERPINE1, whose expression is induced by TGF-β, provides a control for the efficacy of the CHX and Act D.

DOI: https://doi.org/10.7554/eLife.31756.003

The following source data and figure supplements are available for figure 1:

**Figure supplement 1.** SMAD1 phosphorylation kinetics in response to TGF-β.
DOI: https://doi.org/10.7554/eLife.31756.004

**Figure supplements 1—Source data 1.** Source data for qPCRs (panel B).
DOI: https://doi.org/10.7554/eLife.31756.005

**Figure supplement 2.** SMAD1 is efficiently phosphorylated by ACVR1 and BMPR1A, but poorly phosphorylated by TGFBR1.
DOI: https://doi.org/10.7554/eLife.31756.006

Acute TGF-β stimulation results in the rapid internalization of the receptors, which is sufficient to deplete almost all of the type II receptor TGFBR2 from the cell surface (*Vizán et al., 2013*). As a result, cells are refractory to further acute TGF-β stimulation, read out by SMAD2 phosphorylation (*Vizán et al., 2013*). Cells in this refractory state were also unable to induce SMAD1/5 phosphorylation in response to TGF-β, although they remained responsive to BMP4 (*Figure 1C*, *Figure 1—figure supplement 1D*). This suggested that TGFBR2 is required for TGF-β-induced SMAD1/5 activation.

## TGF-β-induced SMAD1/5 phosphorylation requires the kinase activity of two different type I receptors

The distinct kinetics of TGF-β-induced SMAD1/5 phosphorylation compared with SMAD2/3 phosphorylation suggested that different receptor complexes are likely involved. To explore this further, we used combinations of well-characterized small molecule inhibitors of the type I receptor kinases in MDA-MB-231 cells. SB-431542, a selective TGFBR1/ACVR1B/ACVR1C inhibitor (*Inman et al., 2002*), completely inhibited the phosphorylation of both SMAD1/5 and SMAD2 in response to TGF-β when used at 10 μM (*Figure 1D*), indicating that the kinase activity of TGFBR1 is essential for TGF-β-induced SMAD1/5 phosphorylation. Interestingly, a 40-fold lower dose also substantially inhibited SMAD1/5 phosphorylation, whilst having less effect on SMAD2 phosphorylation (*Figure 1D*). TGF-β-induced SMAD1/5 phosphorylation was also partially inhibited by the BMP type I receptor inhibitor LDN-193189 (*Cuny et al., 2008*) (*Figure 1D*), establishing a requirement for a member of this class of type I receptors. Strikingly, the combination of the low-dose SB-431542 and LDN-193189 completely inhibited TGF-β-dependent SMAD1/5 phosphorylation, without affecting phosphorylation of SMAD2 (*Figure 1D*). Analogous results were obtained in NMuMG cells (*Figure 1—figure supplement 1E*).

We conclude that the kinase activity of both classes of type I receptor is required for maximal SMAD1/5 phosphorylation downstream of TGF-β. Taking these results together with the receptor expression profiles of these cells and receptor knockdown experiments (*Daly et al., 2008*), we deduce that the receptors involved are TGFBR1, a canonical BMP type I receptor (ACVR1 and/or BMPR1A) and TGFBR2.

## SMAD1 is primarily phosphorylated by ACVR1

We next used an in vitro approach to explore why TGF-β-induced phosphorylation of SMAD1 requires two different type I receptors. We focused on ACVR1 as a representative of the BMP type I receptor class, as it is the most homologous to ACVRL1 that responds to TGF-β in endothelial cells (*Chen and Massagué, 1999*). Moreover, in some cell types, knockdown of ACVR1 was sufficient to block TGF-β-induced pSMAD1/5 (*Daly et al., 2008*).

SMAD1 is known to be a poor substrate for TGFBR1 in vivo (*Kretzschmar et al., 1997*; *Hoodless et al., 1996*). We demonstrated that SMAD1 is also a poor substrate for TGFBR1 in vitro, although it is efficiently phosphorylated by both ACVR1 and BMPR1A as expected (*Figure 1—figure supplement 2A,B*). As a control, we showed that TGFBR1 could potently phosphorylate SMAD2, and surprisingly, ACVR1 was also able to phosphorylate SMAD2 (*Figure 1—figure supplement 2A, B*).

Given that SMAD1 is a poor substrate for TGFBR1, it is intriguing that the kinase activity of TGFBR1 is essential for TGF-β-induced SMAD1 phosphorylation. We hypothesized that TGFBR1 might catalyze a priming phosphorylation on SMAD1, which then serves as a substrate for ACVR1, or *vice versa*. To address this, we mapped the sites phosphorylated by ACVR1 on full length SMAD1. We identified three species of C-terminal SMAD1 phosphorylation by ACVR1 – a dually phosphorylated S[pS]V[pS] and the singly phosphorylated [pS]SVS and S[pS]VS (*Figure 1—figure supplement 2C*). From this it was clear that ACVR1 could phosphorylate both serines in the critical SVS motif and we deduced that the order of phosphorylation is the penultimate serine of the motif, followed by the terminal one. Moreover, if the preceding serine was phosphorylated, it prevented the phosphorylation of the other sites.

Taking all these results together, we conclude that in response to TGF-β, the receptor kinase that phosphorylates SMAD1 is ACVR1 and not TGFBR1, and it does so on both serines in the SVS motif in a defined order.

## ACVR1 is activated by TGFBR1 in vitro and in vivo

The absence of a role for the TGFBR1 kinase activity in phosphorylating SMAD1 left open the question of why it is required in vivo for TGF-β-induced SMAD1/5 phosphorylation. We postulated that it might be necessary for ACVR1 activation, and therefore investigated whether TGFBR1 could directly phosphorylate ACVR1. Both TGFBR1 and ACVR1 exhibit significant autophosphorylation activity in vitro, which was inhibited by SB-505124 (another more potent TGFBR1 inhibitor; *DaCosta Byfield et al., 2004*) and LDN-193189, respectively (*Figure 2A*). Crucially, when TGFBR1 and ACVR1 were co-incubated, ACVR1 was phosphorylated, even in the presence of LDN-193189, indicating that ACVR1 *is* a substrate of TGFBR1 (*Figure 2A*).

To determine whether TGFBR1 could activate ACVR1 in vivo, we used an optogenetic approach. To this end, we fused the light-oxygen-voltage (LOV) domain of aureochrome1 from *Vaucheria frigida*, which dimerizes upon blue light stimulation (*Sako et al., 2016*), to the C-terminal ends of the intracellular domains of a constitutively-activated TGFBR1 (mutation T204D) (*Wieser et al., 1995*) and of wild-type ACVR1, along with an N-terminal myristoylation motif to anchor them to the plasma membrane (*Figure 2B*; *Supplementary file 1* and *2*). We refer to these constructs as Opto-TGFBR1* and Opto-ACVR1, respectively. We tested their ability, alone or in combination, to induce phosphorylation of SMAD1/5 in NIH-3T3 cells co-transfected with FLAG-SMAD1 to increase the range of the assay. Transfection of the Opto-ACVR1 alone resulted in no phosphorylation of co-transfected FLAG-SMAD1, either in the absence or presence of blue light. However, when Opto-ACVR1 and Opto-TGFBR1* were co-transfected, a robust light-inducible phosphorylation of FLAG-SMAD1 was observed (*Figure 2C*). Importantly, this was inhibited by both SB-505124 and LDN-193189, confirming the involvement of both receptors (*Figure 2D*). This directly demonstrates that TGFBR1 can activate ACVR1 in vivo. As a control, we showed that Opto-TGFBR1* phosphorylated co-expressed GFP-SMAD3 in the presence of light, which was inhibited by SB-505124, but to a much lesser extent by LDN-193189 (*Figure 2E*). As a further control to ensure that the activation of ACVR1 by TGFBR1 required the kinase activity of the latter, we made a kinase-dead version of Opto-TGFBR1. This construct was unable to induce the activity of ACVR1 in a light-inducible manner and was also unable to induce phosphorylation of GFP-SMAD3 (*Figure 2—figure supplement 1*).

To confirm that the light-inducible phosphorylation of FLAG-SMAD1 observed with the combination of Opto-ACVR1 and Opto-TGFBR1* genuinely resulted from activation of Opto-ACVR1 by Opto-TGFBR1*, we generated a mutant version of Opto-ACVR1, in which the serines and threonines of the GS domain were mutated to alanine and valine, respectively. Since phosphorylation of these serines and threonines is required for type I receptor activation, we would expect this mutant to be uninducible (*Wieser et al., 1995*). Indeed, we found that light-inducible phosphorylation of FLAG-SMAD1 was inhibited when this GS domain mutant of Opto-ACVR1 was used instead of the wild-type Opto-ACVR1 (*Figure 2F,G*).

We therefore conclude that the requirement of the kinase activity of both TGFBR1 and ACVR1 for TGF-β-induced phosphorylation of SMAD1/5 reflects a requirement for activation of ACVR1 by TGFBR1 through phosphorylation of the ACVR1 GS domain.

## TGF-β leads to clustering of ACVR1 and TGFBR1

Having shown that both type I receptors are required, we next tested whether they were components of the same tetrameric receptor complex, or whether they resided in separate receptor complexes that clustered at the cell membrane in response to ligand stimulation (compare model I and model II, *Figure 3A*). To distinguish between these possibilities, we used previously published recombinant versions of TGF-β3, designated WW and WD (*Huang et al., 2011*). TGF-β3^WW, the wild-type TGF-β3 dimer, is composed of two identical monomeric TGF-β3 subunits, whereas TGF-β3^WD contains one wild-type subunit of TGF-β3 and one mutated subunit that cannot bind to either TGFBR2 or TGFBR1 (*Huang et al., 2011*). Thus, while the TGF-β3^WW ligand engages two type II:type I pairs in the tetrameric complex, the TGF-β3^WD ligand can only engage one pair. In addition, TGF-β3^WW does not bind ACVR1, and by inference, neither does TGF-β3^WD (data not shown). It was previously demonstrated that TGF-β3^WD binding to a single type II:type I receptor pair is sufficient to induce phosphorylation of SMAD3 (*Huang et al., 2011*). We therefore reasoned that if model I was correct then only TGF-β3^WW would induce phosphorylation of SMAD1/5, as the heterotetrameric complex would not be able to be assembled by TGF-β3^WD. If model II was correct, however,

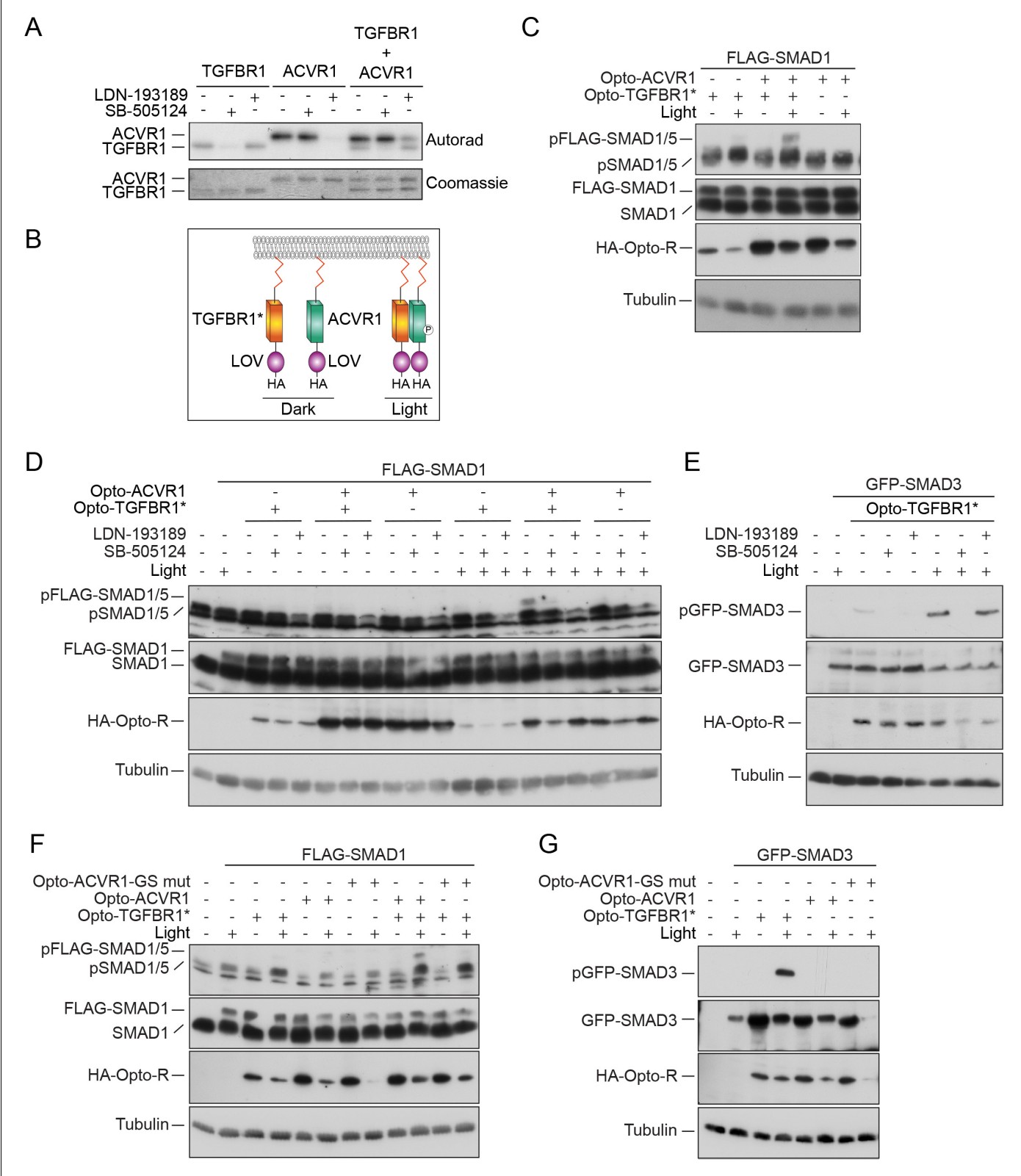

**Figure 2.** ACVR1 is activated by TGFBR1 in vitro and in vivo. (**A**) The kinase domains of TGFBR1 and ACVR1 were analyzed alone or together in an in vitro kinase reaction. SB-505124 and LDN-193189 were included as shown to inhibit the activity of TGFBR1 and ACVR1, respectively. The autoradiograph is shown in the top panel, with the Coomassie-stained gel below as a loading control. (**B**) Schematic to show the domain organization of the Opto receptors. In Opto-TGFBR1* and Opto-ACVR1, the kinase domains of TGFBR1 and ACVR1 are fused to the light-sensitive LOV domain. At

*Figure 2 continued on next page*

*Figure 2 continued*

the N-terminus there is a myristylation domain (indicated by the red zig zag). At the C-terminus there is an HA tag. The kinase domain of TGFBR1 contains the activating mutation T204D. These Opto receptors dimerize in the presence of blue light. (C) NIH-3T3 cells were untransfected or transfected with FLAG-SMAD1 together with either Opto-TGFBR1*, Opto-ACVR1 or both receptors together. Post-transfection, cells were either kept in the dark or exposed to blue light for 1 hr. Whole cell extracts were western blotted using antibodies against pSMAD1/5 (which detects endogenous and FLAG-tagged pSMAD1/5), SMAD1 (which detects endogenous and FLAG-SMAD1), HA (to detect the Opto receptors) and Tubulin as a loading control. (D) NIH-3T3 cells were untransfected or transfected with FLAG-SMAD1 together with either Opto-TGFBR1*, Opto-ACVR1 or both receptors together. Post-transfection, cells were either kept in the dark or exposed to blue light for 1 hr. The inductions were performed in the absence or presence of 0.5 µM LDN-193189 or 50 µM SB-505124 as indicated. Whole cell extracts were blotted as in (C). (E) The experimental set up was as in (D) except that GFP-SMAD3 was used instead of FLAG-SMAD1 to assess the activity of Opto-TGFBR1*. (F) As in (C), except that an ACVR1 mutant in which all the threonines and serines of the GS domain were mutated to valine or alanine respectively, was also assayed. (G) As in (F), except that GFP-SMAD3 was used instead of FLAG-SMAD1. Note that in all cases the 1 hr induction with blue light led to reduced levels of the transfected receptors and substrates.

DOI: https://doi.org/10.7554/eLife.31756.007

The following figure supplement is available for figure 2:

**Figure supplement 1.** Kinase dead Opto-TGFBR1 cannot activate Opto-ACVR1.

DOI: https://doi.org/10.7554/eLife.31756.008

then both TGF-β3$^{WW}$ and TGF-β3$^{WD}$ would be competent to induce pSMAD1/5. Treatment of MDA-MB-231 or NMuMG cells with either TGF-β3$^{WW}$ or TGF-β3$^{WD}$ led to a dose-dependent increase in both SMAD1 and SMAD2 phosphorylation (*Figure 3B*; *Figure 3—figure supplement 1*). Thus, TGF-β stimulation is unlikely to lead to formation of a heterotetrameric complex comprising TGFBR2/TGFBR1/ACVR1, but instead, leads to the formation of a higher order receptor cluster at the cell surface that includes TGFBR2/TGFBR1 complexes and ACVR1.

## TGF-β induces ACVR1 activation in vivo in a TGFBR1-dependent manner

To obtain direct evidence that TGF-β activates ACVR1, we generated an ACVR1 biosensor that fluoresces when activated. In this construct, ACVR1 is fused to the conformation-sensitive circularly permutated yellow fluorescent protein (cpYFP) core of the InversePericam Ca$^{2+}$ sensor and FKBP1A (formerly FKBP12) to make ACVR1-InversePericam-FKBP1A (ACVR1-IPF) (*Michel et al., 2011*). When the receptor is inactive, the FKBP1A moiety binds to the GS domain of the receptor, which suppresses cpYFP fluorescence. Upon ligand induction, phosphorylation of the GS domain releases FKBP1A, allowing the cpYFP to adopt a fluorescent conformation (*Michel et al., 2011*). We first showed that ACVR1-IPF is functional in that it is able to induce phosphorylation of SMAD1/5 when overexpressed (*Figure 4—figure supplement 1A*). We then stably expressed this biosensor in a number of cell lines (*Figure 4—figure supplement 1B*). In the polarized epithelial cell line, MDCKII and in NIH-3T3 fibroblasts, ACVR1-IPF is readily detectable at the cell membrane, as well as in internal structures, and had no adverse effect on the inducibility of these cells in response to TGF-β or BMP4 (*Figure 4—figure supplement 1B–D*). As a control, we showed that ACVR1-IPF was activated in response to FK506 which binds FKBP1A and releases it from the GS domain of ACVR1 (*Wang et al., 1994*) (*Figure 4—figure supplement 1E*). Treatment of the MDCKII ACVR1-IPF cells with TGF-β resulted in a significant increase in fluorescence that was inhibited by SB-431542 (*Figure 4A and B*; *Videos 1–3*). Furthermore, using flow cytometry for a more quantitative approach we demonstrated that the TGF-β-induced increase in fluorescence was blocked by both SB-431542 and a TGF-β neutralizing antibody and was independent of BMP signaling, as it was unaffected by the BMP antagonist Noggin (*Figure 4C*). Similarly, TGF-β also activated ACVR1 in NIH-3T3 ACVR1-IPF cells in a TGFBR1-dependent manner (*Figure 4—figure supplement 1F,G*; *Videos 4–6*).

## Mapping the binding sites on chromatin for TGF-β-induced pSMAD1/5 reveals that *ID* genes are major transcriptional targets of this pathway

Although the existence of TGF-β-induced pSMAD1/5 has been known for some time, its transcriptional role has never been addressed. Earlier experiments had suggested that TGF-β-induced pSMAD1/5 could only be found in complex with pSMAD2/3 (*Daly et al., 2008*), but using optimized immunoprecipitation conditions it was clear that TGF-β-induced pSMAD1/5 can also be part of pSMAD1/5–SMAD4 complexes (*Figure 5A*). We therefore used chromatin immunoprecipitation

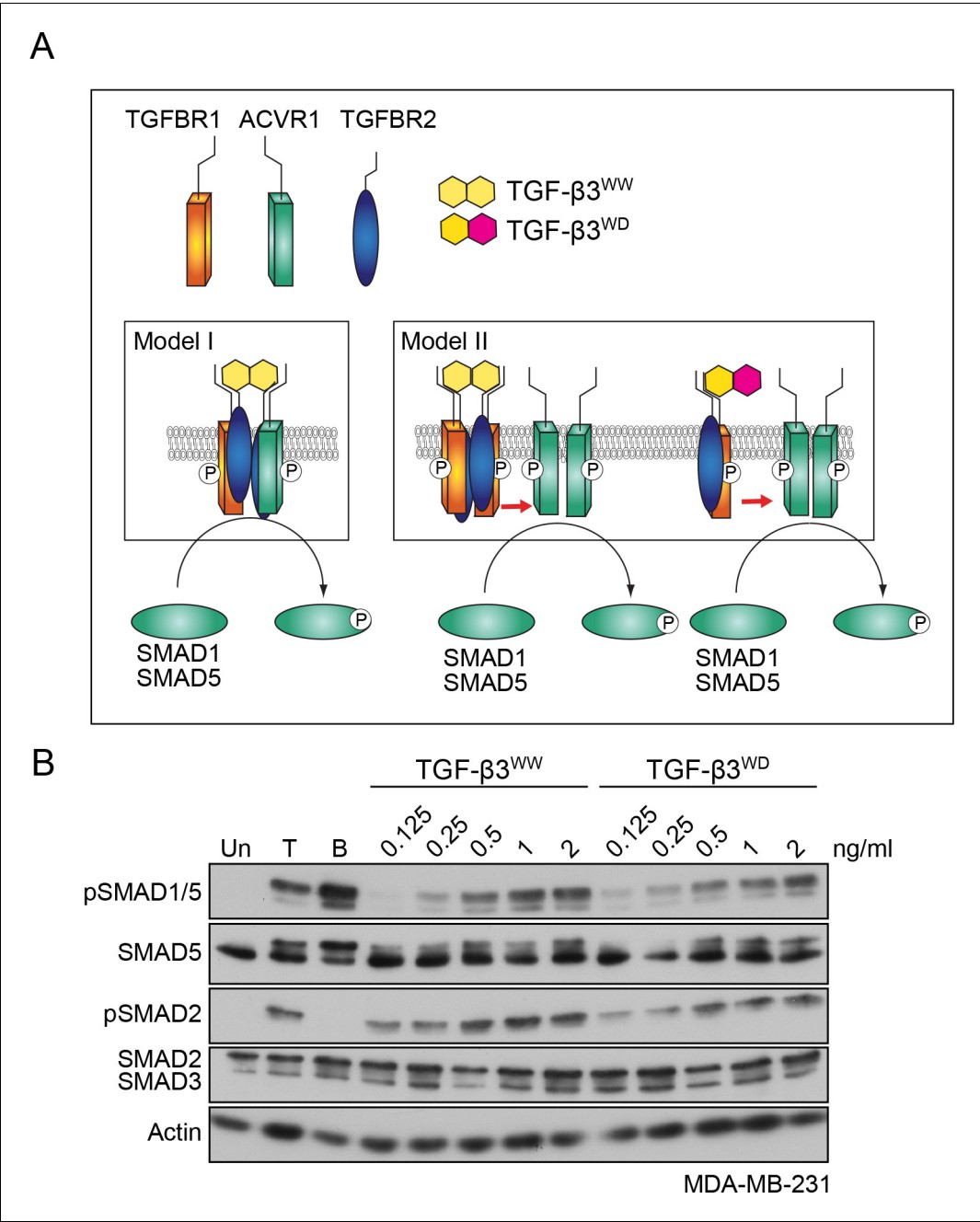

**Figure 3.** TGFBR1 and ACVR1 are present in distinct receptor complexes upon TGF-β stimulation. (**A**) Alternative models of receptor clustering mediated by TGF-β3 derivatives capable of interacting with two pairs (TGF-β3^WW) or one pair (TGF-β3^WD) of type II:type I receptors. If an obligate heterotetramer of two type I:type II pairs is required for SMAD1/5 phosphorylation (Model I), then only TGF-β3^WW would lead to SMAD1/5 phosphorylation. If TGF-β induces higher order receptor clustering at the cell surface (Model II), then both TGF-β3^WW and TGF-β3^WD would lead to SMAD1/5 phosphorylation. (**B**) MDA-MB-231 cells were treated with different concentrations of TGF-β3^WW or TGF-β3^WD for 1 hr as indicated. As a control, cells were either untreated (Un) or treated with TGF-β1 (T) or BMP4 (B) for 1 hr. Whole cell lysates were western blotted using the antibodies shown.

DOI: https://doi.org/10.7554/eLife.31756.009

The following figure supplement is available for figure 3:

**Figure supplement 1.** NMuMG cells respond to both TGF-β3^WW and TGF-β3^WD.

DOI: https://doi.org/10.7554/eLife.31756.010

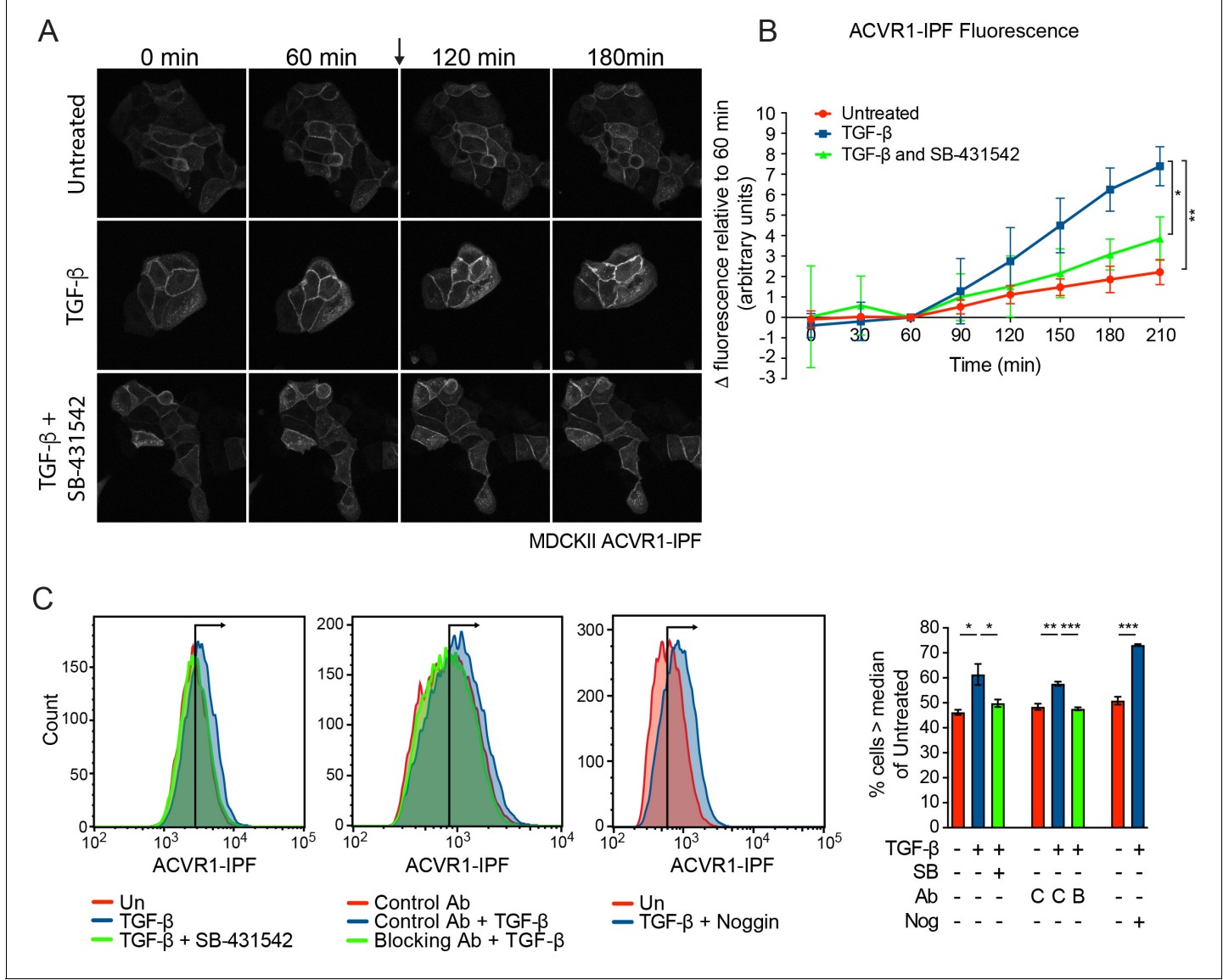

**Figure 4.** ACVR1 is activated by TGF-β in a TGFBR1-dependent manner. (**A and B**) MDCKII ACVR1-IPF cells were imaged at 15 min intervals for 60 min before the addition (arrow) of either media alone or media containing TGF-β ± 10 μM SB-431542 for a further 150 min. The panels in (**A**) are stills of the maximum intensity projections at the times shown. The quantifications are shown in (**B**). The fluorescence at the 60 min time point was taken as the reference that was subtracted from all other time points. Data presented are the mean ± SD of three independent fields. Statistical significance is shown for the indicated pairs of conditions at the 210 min time point. (**C**) Fluorescence in MDCKII ACVR1-IPF cells assayed by flow cytometry 24 hr after treatment. Each panel shows an overlay of the indicated treatment conditions. The black line indicates the median of the untreated (Un) sample. Quantifications are shown on the right. For each group, the percentage of cells greater than the median fluorescence intensity of the untreated sample was quantified. Data are the mean ± SEM of three independent experiments. SB, SB-431542 at 10 μM; Ab, antibody; Nog, noggin; C, control antibody; B, blocking antibody.

DOI: https://doi.org/10.7554/eLife.31756.011

The following source data and figure supplements are available for figure 4:

**Source data 1.** Source data for ACVR1-IPF fluorescence (panel B).
DOI: https://doi.org/10.7554/eLife.31756.016
**Source data 2.** Source data for ACVR1-IPF fluorescence by flow cytometry (panel C).
DOI: https://doi.org/10.7554/eLife.31756.017
**Figure supplement 1.** Characterization of cells stably transfected with the ACVR1-IPF.
DOI: https://doi.org/10.7554/eLife.31756.012
**Figure supplement 1—source data 1.** Source data for ACVR1-IPF fluorescence by flow cytometry (panel E).

*Figure 4 continued on next page*

*Figure 4 continued*

DOI: https://doi.org/10.7554/eLife.31756.013

**Figure supplement 1—source data 2.** Source data for ACVR1-IPF fluorescence (panel F).

DOI: https://doi.org/10.7554/eLife.31756.014

**Figure supplement 1—source data 3.** Source data for ACVR1-IPF fluorescence by flow cytometry (panel G).

DOI: https://doi.org/10.7554/eLife.31756.015

sequencing (ChIP-seq) for pSMAD1/5 to explore where in the genome pSMAD1/5 binds in response to TGF-β. We also wanted to determine which SMAD complexes were primarily responsible for regulating transcription in addition to the canonical pSMAD2/3–SMAD4 complexes (*Figure 5A*).

ChIP-seq in MDA-MB-231 cells for pSMAD1/5 and SMAD3 (as a control) resulted in 2378 pSMAD1/5 peaks and 2440 SMAD3 peaks identified in response to TGF-β after filtering (*Figure 5— source data 1*, sheet 1). The majority of the pSMAD1/5 peaks (2287) were also bound by SMAD3. To identify binding sites preferentially bound by pSMAD1/5 versus SMAD3 we calculated the ratio of the number of tags in the pSMAD1/5 peaks versus the SMAD3 peaks, and focused on the 100 peaks with the highest pSMAD1/5:SMAD3 tag ratio (*Figure 5—source data 1*, sheet 2). Interrogating the nearest genes to these peaks we found a significant enrichment of both TGF-β and BMP target genes (*Figure 5—source data 1*, sheets 2 and 3). Strikingly, 8 of the top 10 peaks flanked known BMP target genes (*ID1, ID3, ID4, ATOH8, BIRC3*) (*Figure 5B*; *Figure 5—figure supplement 1A*; *Figure 5—source data 1*, sheet 2) (*Grönroos et al., 2012*). In contrast, classical TGF-β target genes like *JUNB, BHLHE40, PMEPA1, SERPINE1* (*Levy and Hill, 2005*) were not in this top 100 list, but were amongst those with the highest enrichment for SMAD3 (*Figure 5B*; *Figure 5—figure supplement 1A*; *Figure 5—source data 1*, sheet 1). Using ChIP-qPCR, we validated these different binding patterns (*Figure 5C*; *Figure 5—figure supplement 1A*). For pSMAD1/5, the binding in response to TGF-β was transient, peaking at 1 hr and thereafter decreasing, whilst SMAD3 binding at *JUNB* and *PMEPA1* was sustained. A subset of the peaks were also validated in BT-549 cells (*Figure 5—figure supplement 1B*).

We performed motif enrichment analyses on the top 50 and 100 peaks with the highest pSMAD1/5:SMAD3 tag ratio. In both cases, a SMAD1/5 binding motif GGCGCC was found (*Figure 5D and E*; *Figure 5—figure supplement 1C*) (*Gaarenstroom and Hill, 2014*). In addition, in the top 50 peaks the composite SMAD1/5–SMAD4 site was clearly identified (GGCGCC(N$_5$)GTCT) (*Gaarenstroom and Hill, 2014*; *Morikawa et al., 2011*) (*Figure 5—figure supplement 1C*), with a slightly more degenerate version being present in the top 100 peaks (*Figure 5D*). This strongly suggests that TGF-β-induced SMAD1/5–SMAD4 complexes are responsible for regulating the genes with the highest enrichment of pSMAD1/5.

The enrichment of pSMAD1/5 on the *ID* genes in response to TGF-β suggests that they are *bona fide* target genes of this arm of TGF-β signaling. We confirmed this using siRNAs to deplete specific SMADs. TGF-β induction of *ID1* and *ID3* in MDA-MB-231 cells depended on SMAD1/5 and SMAD4, but not SMAD3 (*Figure 5—figure supplement 2A and B*). In contrast, the induction of *JUNB* required SMAD3 and SMAD4, but was independent of SMAD1/5 (*Figure 5—figure supplement 2A and B*). We further corroborated these observations using the drug dosing strategy that selectively inhibits SMAD1/5 phosphorylation in response to TGF-β (*Figure 1D*). The combination of low-dose SB-431542 and LDN-193189 greatly decreased *ID* gene induction without impacting on the induction of *JUNB* in both MDA-MB-231 and NMuMG cells (*Figure 5—figure supplement 2C and D*). The induction of target gene expression was also examined after treatment of cells with TGF-β3$^{WW}$ or TGF-β3$^{WD}$. As expected both TGF-β ligands induced the expression of the *ID*s and *JUNB* (*Figure 5—figure supplement 2E*).

The results in this section reveal that pSMAD1/5–SMAD4 complexes formed in response to TGF-β are responsible for regulating the genes with the highest enrichment of pSMAD1/5, and that the *ID*s are major early downstream targets.

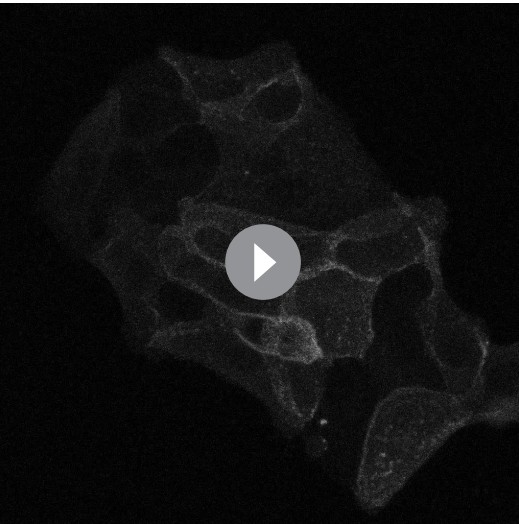

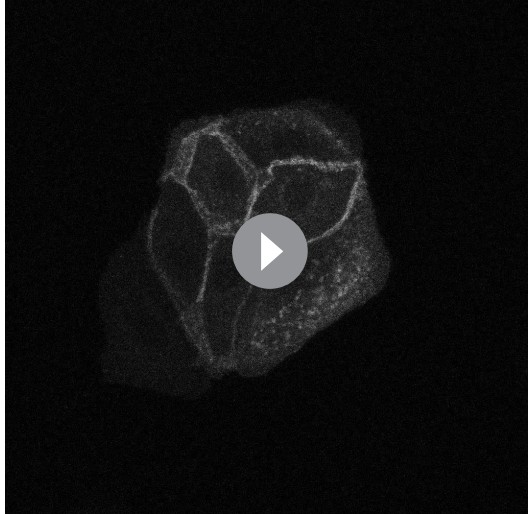

**Video 1.** Fluorescence in MDCKII ACVR1-IPF cells treated with media alone. MDCKII ACVR1-IPF cells were imaged for 1 hr prior to the addition of media alone followed by imaging for a further 2.5 hr. Very little increase in fluorescence was observed over the time course.

DOI: https://doi.org/10.7554/eLife.31756.018

**Video 2.** Fluorescence in MDCKII ACVR1-IPF cells treated with TGF-β. MDCKII ACVR1-IPF cells were imaged for 1 hr prior to the addition of 2 ng/ml TGF-β followed by imaging for a further 2.5 hr. Significant increase in fluorescence was observed over the time course with intracellular puncta of fluorescence becoming more evident over time.

DOI: https://doi.org/10.7554/eLife.31756.019

## The SMAD1/5 arm of TGF-β signaling is required for TGF-β-induced EMT

The ID proteins have been implicated in many processes involved in oncogenesis (*Lasorella et al., 2014*), and importantly, ID1 was shown to be upregulated by TGF-β in tumor cells isolated from pathological pleural fluids from patients with ER− and ER+ metastatic breast cancer, and also in patient-derived glioblastomas (*Anido et al., 2010*; *Padua et al., 2008*). Since we have now shown that the pSMAD1/5 arm of TGF-β signaling is responsible for TGF-β-induced *ID1* induction, this prompted us to explore further the biological relevance of the pSMAD1/5 arm of TGF-β signaling in oncogenic processes, and to gain a comprehensive view on the relative contribution of this arm of signaling to longer term TGF-β responses. We decided to focus on the process of EMT, as this is a key step in tumorigenesis that confers a migratory phenotype, acquisition of stem cell properties and resistance to chemotherapeutic agents (*Ye and Weinberg, 2015*). For these studies, we primarily used the NMuMG cell model, as we have shown above that these cells show a robust phosphorylation of SMAD1/5 in response to TGF-β and are well known to undergo a TGF-β-induced EMT within 48 hr (*Piek et al., 1999*).

CRISPR/Cas9 was used to generate clones of NMuMG cells deleted for SMAD1 and SMAD5 (*Figure 6A*; *Figure 6—figure supplement 1A–C*). We compared the TGF-β-induced transcriptome at 48 hr of the parental clone with one deleted for SMAD1/5 using RNA-sequencing (RNA-seq). Of the 5798 genes that are significantly up- or down-regulated by TGF-β in this time frame we found that approximately a quarter (1398) were dependent on the SMAD1/5 branch of signaling (see Materials and methods for the cut-offs used) (*Figure 6—source data 1*, sheets 1 and 2). This demonstrates that this arm of TGF-β signaling plays a crucial role in long term downstream transcription responses. To corroborate the RNA-seq results, we validated a subset of them by qPCR, measuring levels of mRNA over time in response to TGF-β (*Figure 6—figure supplement 2*).

Gene set enrichment analysis revealed that the TGF-β target genes that depend on this arm of signaling were involved in processes such as regulation of the cytoskeleton, focal adhesions, adherens and tight junctions, as well as EMT (*Figure 6—source data 1*, sheet 3). We therefore next investigated whether TGF-β-induced EMT required SMAD1/5 signaling. Using delocalization of the adherens junction marker

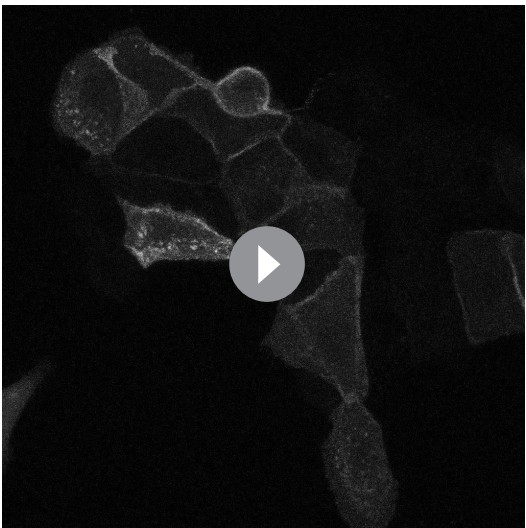

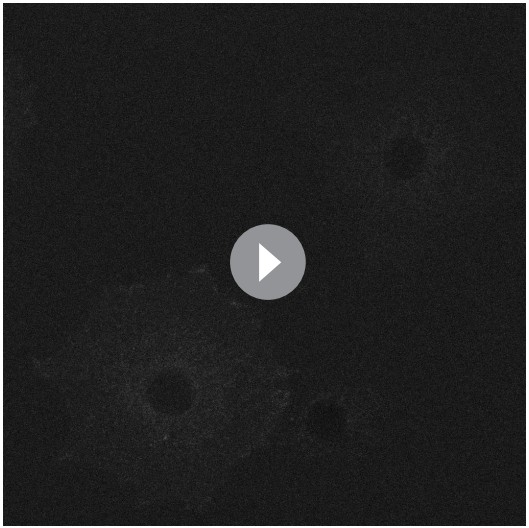

**Video 3.** Fluorescence in MDCKII ACVR1-IPF cells treated with TGF-β and SB-431542. MDCKII ACVR1-IPF cells were imaged for 1 hr prior to the addition of 2 ng/ml TGF-β + 10 μM SB-431542 to the cells followed by imaging for a further 2.5 hr. Very little increase in fluorescence was observed over the time course.
DOI: https://doi.org/10.7554/eLife.31756.020

**Video 4.** Fluorescence in NIH-3T3 ACVR1-IPF cells treated with media alone. NIH-3T3 ACVR1-IPF cells were imaged for 3.5 hr after the addition of media alone. A modest and gradual increase in fluorescence was observed over the time course.
DOI: https://doi.org/10.7554/eLife.31756.021

CDH1 (also called E-Cadherin) together with loss of the tight junction marker TJP1 (also called ZO-1) as a measure of EMT, we could readily demonstrate that signaling through SMAD1/5 was crucial for this process in two separate ΔSMAD1/5 clones (*Figure 6B*; *Figure 6—figure supplement 1D*). In addition, we observed that two mesenchymal markers, *Acta2* (also called smooth muscle actin) and *Fn1* were more weakly induced in the ΔSMAD1/5 clone compared with the wild-type (*Figure 6—figure supplement 2*). We also used an siRNA knockdown approach, and showed that EMT was dependent on SMAD1/5, SMAD4 and SMAD3, but independent of SMAD2 (*Figure 6—figure supplement 3A and B*). Furthermore, treatment of the cells with the BMP type I receptor inhibitor, LDN-193189 also inhibited EMT either alone or when combined with low-dose SB-431542 which we have shown is sufficient to inhibit TGF-β-induced SMAD1/5 signaling, but not signaling through SMAD2/3 (*Figure 6C and D*; *Figure 6—figure supplement 3C*). Moreover, DMH1, another BMP type I receptor inhibitor, had a similar effect (*Figure 6—figure supplement 3C and D*). Finally, to confirm that the dependence of TGF-β-induced EMT on SMAD1/5 signaling was not unique to NMuMG cells, we used another mouse mammary cell line, EpRas that also undergoes a TGF-β-induced EMT (*Daly et al., 2010*; *Grünert et al., 2003*). SMAD1/5 signaling in this line was also essential for EMT (*Figure 6E and F*). Thus, we conclude that TGF-β-induced EMT requires the SMAD1/5 arm of the signaling pathway, as well as the canonical pathway through SMAD3.

Taking our ChIP-seq and RNA-seq analyses together, we found that the *ID* genes are major early transcriptional targets of the SMAD1/5 arm of the TGF-β pathway. Of these, ID1 was the prominent family member up-regulated by TGF-β in NMuMGs (*Figure 7—figure supplement 1A*). We hypothesized that the dependency on the SMAD1/5 arm of the TGF-β pathway could reflect a requirement of ID1 for EMT. We tested this by knocking down *Id1* with siRNAs, both as a pool and as individual siRNAs and found that cells depleted of ID1 were indeed unable to undergo TGF-β-induced EMT (*Figure 7A and B*; *Figure 7—figure supplement 1B and C*). Thus, we conclude that TGF-β-induced up-regulation of ID1 is essential for EMT.

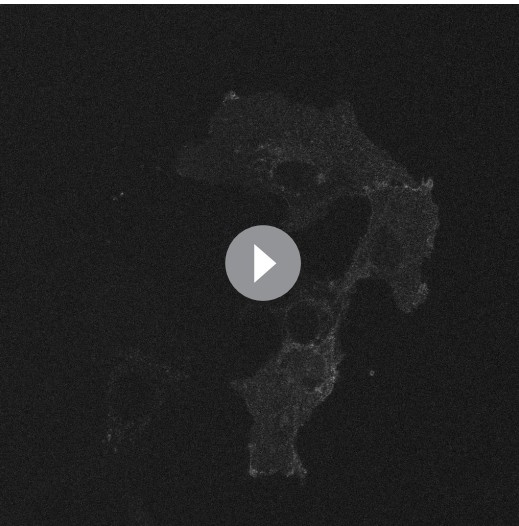 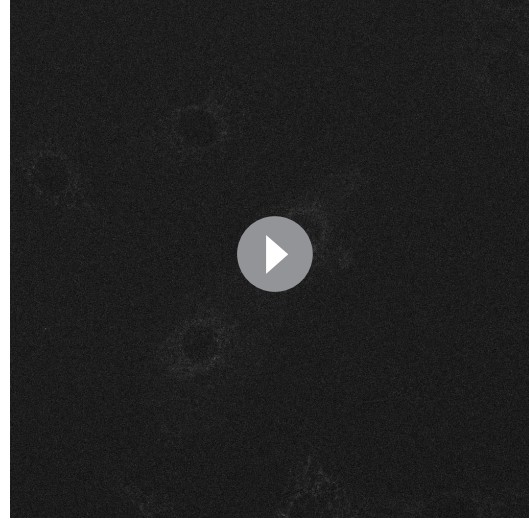

**Video 5.** Fluorescence in NIH-3T3 ACVR1-IPF cells treated with TGF-β. NIH-3T3 ACVR1-IPF cells were imaged for 3.5 hr after the addition of 2 ng/ml TGF-β. A significant increase in fluorescence was observed over the time course with fluorescence becoming more evident on membrane projections and intracellular vesicles over time.
DOI: https://doi.org/10.7554/eLife.31756.022

**Video 6.** Fluorescence in NIH-3T3 ACVR1-IPF cells treated with TGF-β and SB-431542. NIH-3T3 ACVR1-IPF cells were imaged for 3.5 hr after the addition of 2 ng/ml TGF-β + 10 μM SB-431542. A modest and gradual increase in fluorescence was observed over the time course.
DOI: https://doi.org/10.7554/eLife.31756.023

## Discussion

### Combinatorial signaling downstream of TGF-β

Here, we have defined both the mechanism whereby TGF-β induces the phosphorylation of SMAD1/5, and its functional role. We have shown that two type I receptors are required, the canonical TGF-β receptor TGFBR1, and additionally, one of the classical BMP type I receptors, ACVR1. Using in vitro kinase assays, an optogenetic approach and an ACVR1 receptor fluorescent biosensor, we have uncovered a new mechanism for receptor activation whereby one type I receptor activates another. We show that in response to TGF-β, TGFBRI phosphorylates and activates ACVR1, which phosphorylates SMAD1/5. To address the functional significance of this arm of TGF-β signaling, we used genome-wide ChIP-seq and RNA-seq and show that approximately a quarter of the TGF-β-regulated transcriptome is dependent on SMAD1/5, with major early targets being the ID transcriptional regulators. Finally, we have also demonstrated that the SMAD1/5 pathway is essential for TGF-β-induced EMT, and this reflects a requirement for ID1.

Taking these results together with previous work (*Liu et al., 2009*; *Daly et al., 2008*), we propose a model of combinatorial signaling that is essential for the TGF-β cellular program (*Figure 7C*). In most cells tested the induction of pSMAD1/5 is more transient than the pSMAD2/3 induction (*Liu et al., 2009*; *Daly et al., 2008*). Thus, the initial transcriptional program is regulated by both SMAD pathways and is refined at later time points by the SMAD2/3 pathway. Therefore, the full TGF-β-induced transcriptional program requires combinatorial signaling via both SMAD pathways. With respect to the functional relevance of TGF-β-induced SMAD1/5 phosphorylation, we have now shown that complete EMT requires both SMAD pathways. TGF-β-induced anchorage-independent growth, migration and invasion have also been shown to require SMAD1/5 signaling, whilst TGF-β-induced growth arrest is only dependent on SMAD2/3 signaling (*Liu et al., 2009*; *Daly et al., 2008*) (*Figure 7C*).

Since we have now demonstrated that TGF-β induces the formation of SMAD1/5–SMAD4 complexes that regulate canonical BMP target genes, it is important to ask what discriminates TGF-β

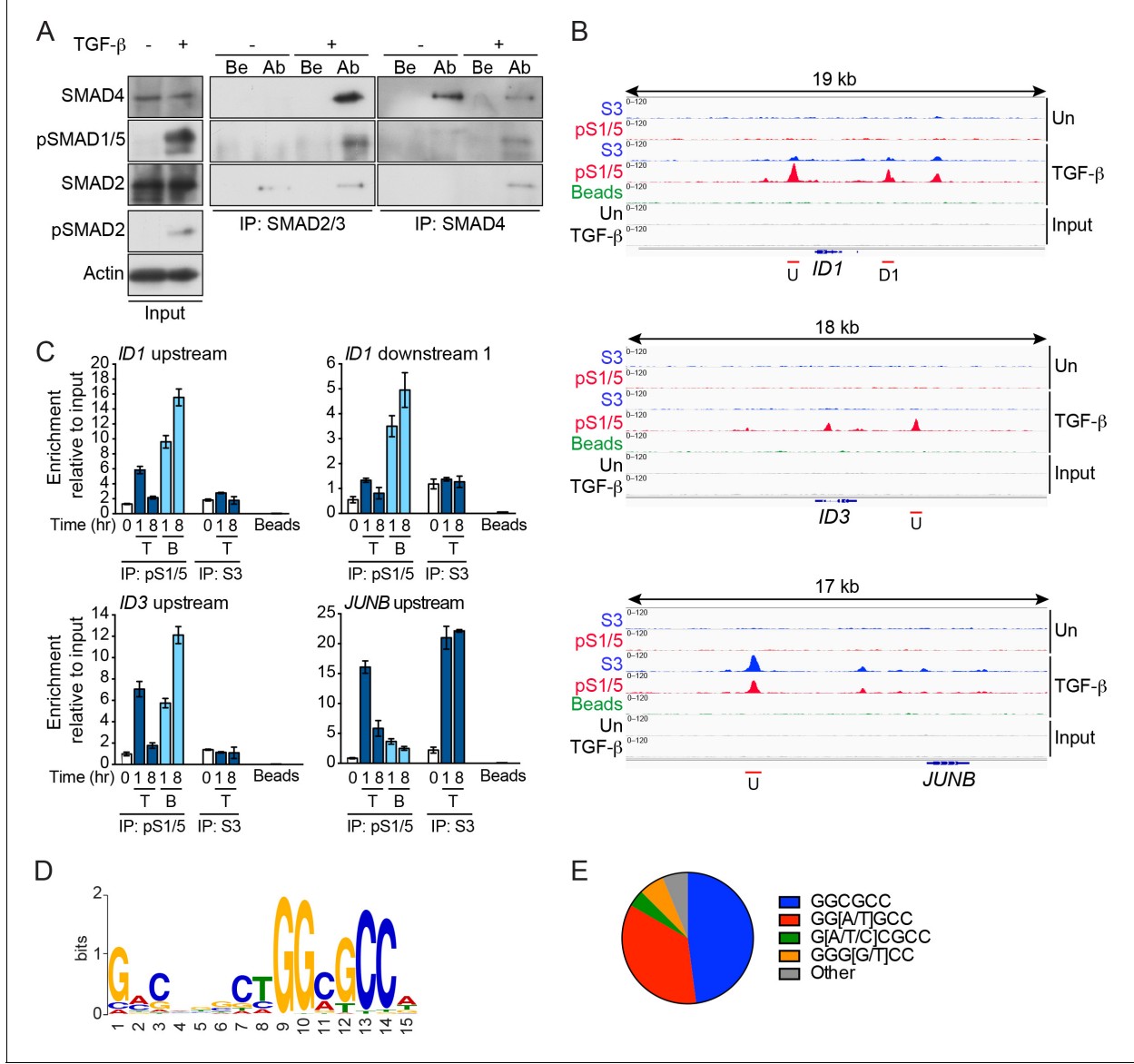

**Figure 5.** pSMAD1/5 is recruited to chromatin in response to TGF-β and is most highly enriched at GGCGCC motifs. (**A**) MDA-MB-231 cells were either untreated (-) or treated with TGF-β (+) for 1 hr. Whole cell extracts were immunoprecipitated (IP) with the antibodies (Ab) indicated or beads alone (Be). The IPs were western blotted using the antibodies shown. Inputs are shown on the left. (**B**) IGV browser displays over the *ID1*, *ID3* and *JUNB* loci after ChIP-Seq of MDA-MB-231 untreated (Un) and TGF-β-treated samples. IPs were performed with antibodies against pSMAD1/5 (pS1/5), SMAD3 (S3) or with beads alone as a negative control. Inputs are also shown. Red lines indicate regions validated in (**C**). U; upstream peak; D1, downstream peak 1. (**C**) Genomic regions were validated by ChIP-qPCR after treatment of MDA-MB-231 cells with TGF-β (T) or BMP4 (B) for the times shown. IPs were as in (**B**). A representative experiment of two performed in triplicate is shown with means ± SD. (**D**) The most enriched motif obtained from a MEME-ChIP analysis of the top 100 pSMAD1/5 peaks. (**E**) Proportion of variants of the GGCGCC motif identified in the top 100 pSMAD1/5 peaks.
DOI: https://doi.org/10.7554/eLife.31756.024

The following source data and figure supplements are available for figure 5:

**Source data 1.** ChIP-seq datasets.
DOI: https://doi.org/10.7554/eLife.31756.033
**Source data 2.** ChIP-PCR data for graphs in panel C.
DOI: https://doi.org/10.7554/eLife.31756.034
**Figure supplement 1.** Chromatin binding of pSMAD1/5 and SMAD3.
DOI: https://doi.org/10.7554/eLife.31756.025
**Figure supplement 1—source data 1.** ChIP-PCR data for graphs in panel A.
DOI: https://doi.org/10.7554/eLife.31756.026
*Figure 5 continued on next page*

*Figure 5 continued*

**Figure supplement 1—source data 2.** ChIP-PCR data for graphs in panel B.

DOI: https://doi.org/10.7554/eLife.31756.027

**Figure supplement 2.** *ID1* and *ID3* are TGF-β-induced target genes that require the pSMAD1/5 signaling arm.

DOI: https://doi.org/10.7554/eLife.31756.028

**Figure supplement 2—source data 1.** qPCR data for graphs in panel B.

DOI: https://doi.org/10.7554/eLife.31756.029

**Figure supplement 2—source data 2.** qPCR data for graphs in panel C.

DOI: https://doi.org/10.7554/eLife.31756.030

**Figure supplement 2—source data 3.** qPCR data for graphs in panel D.

DOI: https://doi.org/10.7554/eLife.31756.031

**Figure supplement 2—source data 4.** qPCR data for graphs in panel E.

DOI: https://doi.org/10.7554/eLife.31756.032

signaling from BMP signaling as it is well known that BMP and TGF-β functional responses are distinct (*Itoh et al., 2014*; *Miyazono et al., 2010*). The answer lies in the combinatorial signaling, and likely also in the signaling dynamics. In contrast to TGF-β, BMP stimulation leads to a sustained phosphorylation of SMAD1/5 in the absence of SMAD2/3 activation (*Grönroos et al., 2012*; *Daly et al., 2008*). As a result, although the gene expression program downstream of BMP shares some common targets with that downstream of TGF-β at early time points, it will be completely distinct at later time points as a result of the sustained SMAD1/5 signaling and the absence of SMAD2/3-driven transcription (*Figure 7C*).

## Receptor requirements for TGF-β-induced SMAD1/5 phosphorylation

We have shown that two classes of type I receptors are necessary for TGF-β-induced SMAD1/5 phosphorylation, the canonical TGF-β receptor, TGFBR1 and one of the BMP type I receptors, of which we have focused on ACVR1. Our results demonstrate that the kinase activity of TGFBR1 is essential for activation of ACVR1, whereas the kinase activity of ACVR1 is necessary to phosphorylate SMAD1/5. Surprisingly, we found that inhibition of TGF-β-induced SMAD1/5 phosphorylation by LDN-193189, which inhibits the BMP type I receptors, is incomplete, even though the same LDN-193189 concentration is sufficient to inhibit BMP-induced SMAD1/5 phosphorylation. This same result was also previously seen when the BMP type I receptor inhibitor dorsomorphin was used (*Daly et al., 2008*). A complete inhibition of TGF-β-induced pSMAD1/5 is achieved by combining LDN-193189 with a sub-optimal dose of SB-431542. This is likely explained by the fact that LDN-193189-inhibited ACVR1 is still able to efficiently recruit SMAD1/5, where it may be inefficiently phosphorylated by TGFBR1, which is sensitive to the sub-optimal dose of SB-431542. The requirement for two distinct type I receptors fits well with what was shown for TGF-β responses in endothelial cells, where ACVLR1 and TGFBR1 were both required (*Goumans et al., 2003*; *Goumans et al., 2002*).

Our optogenetic experiments revealed that activated TGFBR1 phosphorylates and activates ACVR1 in vivo. We previously hypothesized that TGFBR1 and ACVR1 could be in the same receptor complex (*Daly et al., 2008*), but our use of the mutant TGF-β3 ligands here indicated that these two type I receptors are not part of an obligate heterotetrameric receptor, but rather that TGFBR1, activated by TGFBR2 as a result of TGF-β stimulation can phosphorylate and activate ACVR1 in the membrane as a result of receptor clustering (*Figure 7C*). We were surprised to see in our optogenetic experiments that light-induced dimers of the activated kinase domain of TGFBR1 were much more active than the monomeric domains, as this suggests that TGFBR1 is able to autophosphorylate and auto-activate in the absence of type II receptors, if brought into close proximity. In fact, a similar observation was made in early studies using chimeric receptors with the extracellular domain of the erythropoietin receptor and the intracellular domain of constitutively active TGFBR1 (*Luo and Lodish, 1996*). This chimeric receptor could only mediate a growth arrest after stimulation with erythropoietin, indicating that clustering was important for receptor activity in vivo.

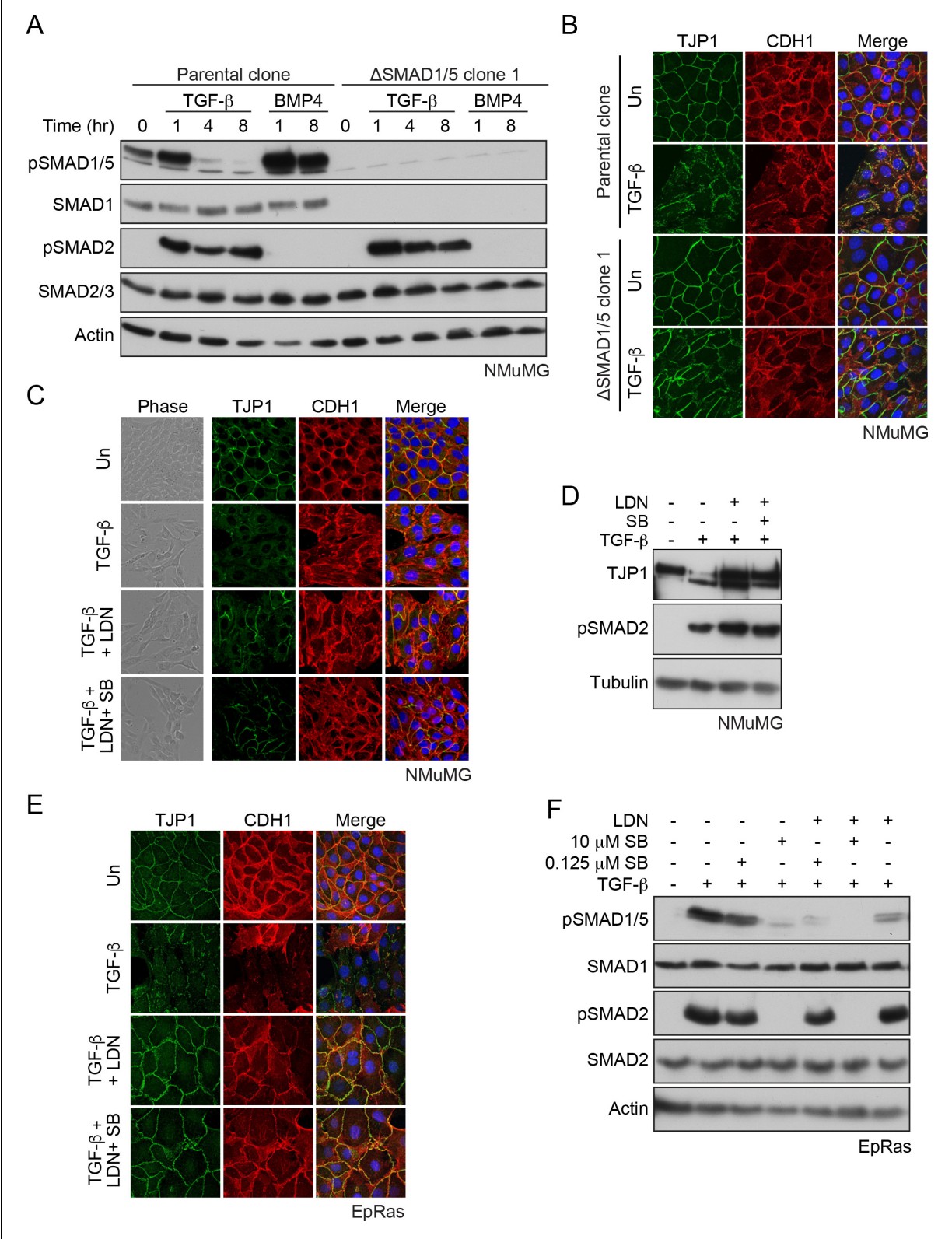

**Figure 6.** SMAD1/5 is required for TGF-β-induced EMT. (**A**) The parental NMuMG clone and the ΔSMAD1/5 clone 1 were treated with TGF-β or BMP4 for the times shown. Whole cell extracts were immunoblotted with the antibodies indicated. (**B**) Parental NMuMG clone and the ΔSMAD1/5 clone 1 cells were left untreated or treated with TGF-β for 48 hr and imaged after indirect immunofluorescence (IF) using antibodies against TJP1 and CDH1. A merge of the two with DAPI in blue is also shown. (**C**) NMuMG cells were left untreated (Un) or treated with TGF-β alone or in combination with 1 μM

*Figure 6 continued on next page*

*Figure 6 continued*

LDN-193189 (LDN) ± 0.125 μM SB-431542 (SB) for 48 hr. Panels show cells imaged under either phase contrast (left panels) or by indirect immunofluorescence (IF) using antibodies against TJP1 and CDH1. A merge of the two with DAPI in blue is also shown. (D) NMuMG cells were left untreated or treated with TGF-β alone or in combination with either 1 μM LDN-193189 ± 0.125 μM SB-431542 for 48 hr. Whole cell lysates were immunoblotted with the indicated antibodies. (E) EpRas cells were left untreated (Un) or treated with TGF-β alone or in combination with 1 μM LDN-193189 (LDN) ± 0.125 μM SB-431542 (SB) for 9 days, then imaged after indirect immunofluorescence (IF) using antibodies against TJP1 and CDH1 or a merge of the two with DAPI in blue. (F) EpRas cells were left untreated or treated with TGF-β for 1 hr alone or with combinations of 1 μM LDN-193189, 0.125 μM SB-431542 or 10 μM SB-431542 as indicated. Whole cell lysates were immunoblotted with the indicated antibodies. In (B), (C) and (E) the indirect IF images are maximum intensity projections of a z-stack in each channel.
DOI: https://doi.org/10.7554/eLife.31756.035

The following source data and figure supplements are available for figure 6:

**Source data 1.** RNA-seq datasets.
DOI: https://doi.org/10.7554/eLife.31756.040
**Figure supplement 1.** Characterization of the NMuMG ΔSMAD1/5 clones.
DOI: https://doi.org/10.7554/eLife.31756.036
**Figure supplement 2.** Validation of SMAD1/5-dependent TGF-β-induced genes.
DOI: https://doi.org/10.7554/eLife.31756.037
**Figure supplement 2—Source data 1.** qPCR data for all graphs shown.
DOI: https://doi.org/10.7554/eLife.31756.038
**Figure supplement 3.** The pSMAD1/5 signaling arm is required for TGF-β-mediated EMT.
DOI: https://doi.org/10.7554/eLife.31756.039

## Dynamics of TGF-β-induced pSMAD1/5 signaling

We and others have observed that in most cell types TGF-β-induced SMAD1/5 phosphorylation is transient compared with SMAD2/3 phosphorylation (*Daly et al., 2008*; *Liu et al., 2009*; *Wrighton et al., 2009*). Using pSMAD2 as a readout, we previously showed that pSMAD2 levels attenuate over time, and remain at a low steady state level that depends on receptors replenishing the cell surface, for as long as ligand is available (*Vizán et al., 2013*). Our demonstration that levels of fluorescence of the ACVR1-IPF biosensor steadily increase over a number of hours indicates that ACVR1 can also be continuously activated for as long as ligand is present. We have shown that the transience of SMAD1/5 phosphorylation requires new protein synthesis, indicating that SMAD1/5 phosphorylation is likely to be actively terminated by an inhibitor induced by the pathway. Given the prolonged activation of ACVR1-IPF in response to ligand, we hypothesize that such an inhibitor is unlikely to target the receptors, but might be a TGF-β-induced phosphatase that targets phosphorylated SMAD1/5 directly. The transience of SMAD1/5 phosphorylation is not a defining characteristic of this arm of TGF-β signaling as BT-549 breast cancer cells exhibit a more sustained response, which is even more pronounced when the cells are grown as spheres. Comparing TGF-β target genes in BT-549s versus MDA-MB-231s where the response is transient, might shed light on the identity of the putative inhibitor.

## TGF-β-induced pSMAD1/5 is transcriptionally active and required for a subset of TGF-β-induced target genes

Our ChIP-seq analysis demonstrates for the first time that TGF-β-induced pSMAD1/5 accumulates in the nucleus and binds to chromatin. These experiments revealed that the peaks with the highest pSMAD1/5 enrichment flanked classical BMP target genes, such as *ID1, ID3* and *ATOH8*. Analysis of the binding sites led us to the discovery that the SMAD complexes responsible for inducing these target genes downstream of TGF-β were pSMAD1/5–SMAD4 complexes. The ChIP-seq analysis also revealed widespread co-binding of pSMAD1/5 and SMAD3, which was surprising. For the classical BMP targets, the ratio of pSMAD1/5:SMAD3 in the peaks was high, whereas at classical TGF-β targets like *JUNB, PMEPA1, SERPINE1* and *BHLHE40* (*Kang et al., 2003*; *Levy and Hill, 2005*), this ratio was less than 1. We do not fully understand the functional significance of the pSMAD1/5 and SMAD3 co-binding. We previously demonstrated that at least in some contexts, pSMAD3–pSMAD1/5 complexes are inhibitory (*Grönroos et al., 2012*), and this is evident in the work presented here for *ID3* induction. However, for *JUNB* we found that knockdown of SMAD1/5 had no effect on TGF-

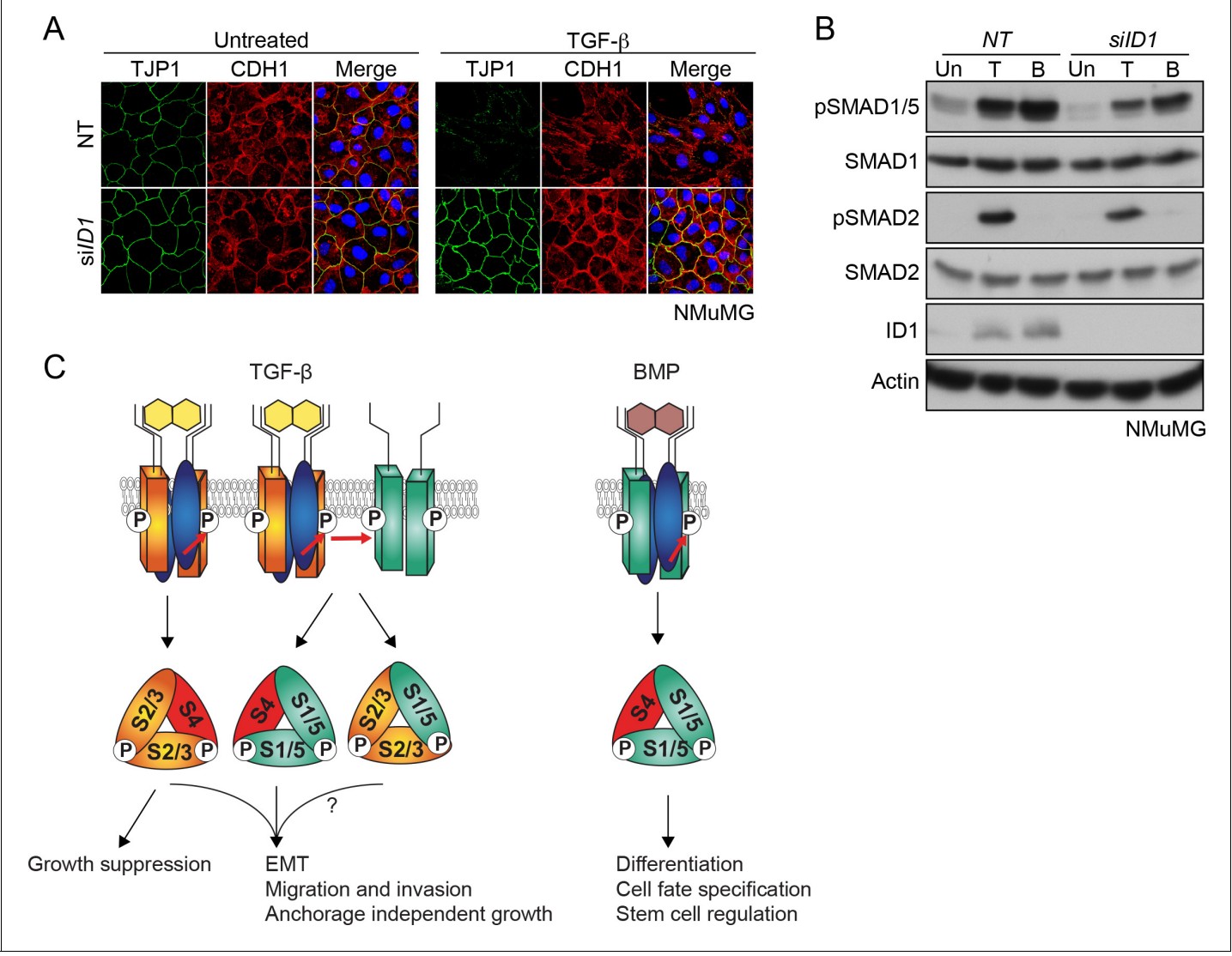

**Figure 7.** TGF-β-induced ID1 via pSMAD1/5 is required for EMT. (**A**) NMuMG cells were transfected with siRNAs against *ID1* or NT control, then left untreated or treated with TGF-β for 24 hr. Cells were imaged after indirect IF with antibodies against TJP1 and CDH1 or a merge of the two with DAPI in blue. All indirect IF images are maximum intensity projections of a z-stack in each channel. (**B**) Western blots to show knockdown efficiency of the *ID1* siRNA. NMuMG cells were treated with TGF-β (T) or BMP4 (B) for 1 hr. (**C**) The model shows combinatorial signaling by TGF-β utilizing complexes containing two different type I receptors. Type II receptors are shown in blue, TGFBR1 in orange and ACVR1 in green as in *Figure 3A*. P denotes phosphorylation. S1/5, SMAD1/5; S2/3, SMAD2/3; S4, SMAD4. The question mark indicates that we do not yet know the function of the mixed R-SMAD complexes in the physiological responses. For discussion, see text.

DOI: https://doi.org/10.7554/eLife.31756.041

The following figure supplement is available for figure 7:

**Figure supplement 1.** TGF-β-induced ID1 is required for EMT.
DOI: https://doi.org/10.7554/eLife.31756.042

β-induced transcription, suggesting that pSMAD1/5 is not contributing to its transcriptional regulation. This may also be true of other genes with a similar pattern of SMAD3/pSMAD1/5 binding.

## TGF-β-induced SMAD1/5 signaling is required for EMT through induction of ID1

We have now shown that SMAD1/5 signaling in response to TGF-β is required for a complete TGF-β-induced EMT in NMuMG cells and in EpRas cells. This accounts for a previously unexplained

observation that overexpression of dominant negative ACVR1 in NMuMGs caused a partial loss of EMT in response to TGF-β (*Miettinen et al., 1994*). In an earlier study using siRNAs we had concluded that the SMAD1/5 arm of the TGF-β pathway was not required for EMT in EpRas cells (*Daly et al., 2008*). The likely explanation for this discrepancy is the poor SMAD1/5 knockdown we achieved in those cells compared with the very effective strategy of inhibiting this arm of TGF-β signaling using the combined small molecule inhibitors that we have employed here.

We have gone on to show that TGF-β-induced ID1 is required for EMT. Importantly, although ID1 is necessary for EMT, it is clearly not sufficient, as BMPs cannot induce EMT in NMuMGs (*Kowanetz et al., 2004*). Consistent with this we have also shown that the SMAD3 pathway is essential for EMT. This arm of the pathway is likely required for the induction of some or all of the so-called EMT-associated transcription factors, most notably SNAI1, SNAI2, ZEB1, ZEB2 and BHLH proteins such as TWIST1 and E47 (now called TCF3), some of which are known direct TGF-β targets (*Peinado et al., 2007*; *Diepenbruck and Christofori, 2016*).

Our finding that EMT depends on TGF-β-induced ID1 expression has implications for the role of SMAD1/5 and the IDs in cancer. The prevailing view is that ID1 is downregulated by TGF-β in non-tumorigenic human epithelial lines, but upregulated by TGF-β in established tumor cell lines, as we have observed here in MDA-MB-231 and BT-549s, and also in patient-derived tumor cells (*Anido et al., 2010*; *Padua et al., 2008*; *Lasorella et al., 2014*). Furthermore, ID proteins are overexpressed in many different tumor types and are implicated in the maintenance of tumor stem cells and for some cancer-related phenotypes (*Lasorella et al., 2014*). ID1 was also found in a lung metastatic gene signature of breast cancer (*Minn et al., 2005*). The role of ID1 in EMT is context dependent. In a recent study of breast cancer, ID1 was shown to be expressed in tumor cells that had already undergone an EMT, and it contributed to the growth of the primary tumor by inducing a stem cell-like phenotype. At the metastatic site, however, TGF-β-induced ID1 was proposed to induce an mesenchymal-to-epithelial transition (MET) by interferring with the activity of TWIST (*Stankic et al., 2013*). In light of our current data it will be important to investigate in what tumor contexts ID1 is required for EMT, and more broadly how the TGF-β–SMAD1/5 pathway contributes to different aspects of tumorigenesis.

## Materials and methods

### Cell line origin, authentication and maintenance

MDA-MB-231 cells were obtained from the ECACC/HPA culture collection, BT-549 cells were obtained from the Francis Crick Institute Cell Services, NMuMG cells were obtained from ATCC, MDCKII cells were obtained from Sigma (UK), NIH-3T3 cells were obtained from Richard Treisman (Francis Crick Institute) and EpRas cells were obtained from Martin Oft and Hartmut Beug (IMP, Vienna). All cell lines have been banked by the Francis Crick Institute Cell Services, and certified negative for mycoplasma. In addition, MDA-MB-231 and BT-549 cells were authenticated using the short tandem repeat profiling, while MDCKII, NIH-3T3 and EpRas cells had species confirmation at the Francis Crick Institute Cell Services. Their identity was also authenticated by confirming that their responses to ligands and their phenotype were consistent with published history.

MDA-MB-231, BT-549, EpRas, NIH-3T3 and MDCKII cells were maintained in Dulbecco's modified Eagle's medium (DMEM) supplemented with 10% fetal calf serum (FCS) and 1% penicillin/streptomycin. NMuMG cells were grown in the same medium, but supplemented with 10 μg/ml insulin. MDA-MB-231 and MDCKII cells were starved overnight in OptiMEM prior to ligand stimulation; NMuMG cells were starved overnight in OptiMEM with 10 μg/ml insulin; NIH-3T3 cells were starved in DMEM with 0.5% FCS. For ligand stimulation experiments, BT-549 cells were plated in the mammosphere culture media (*Dontu et al., 2003*) (MEBM (PromoCell, Germany) with B27 (Thermo Fisher, UK), 20 ng/ml EGF (PeproTech, UK), 20 ng/ml bFGF (PeproTech) and 4 μg/ml heparin (Sigma).

### Ligands and chemicals

All recombinant ligands were reconstituted in 4.4 mM HCl supplemented with 0.1% BSA. Cells were treated with recombinant TGF-β1 (PeproTech, 100–21C; 2 ng/ml), BMP4 (PeproTech, 120-05ET; 20 ng/ml) and Noggin (PeproTech, 250–38; 300 ng/ml). TGF-β3$^{WW}$ and TGF-β3$^{WD}$ were as described (*Huang et al., 2011*). SB-431542 (Tocris, UK) was used at the concentrations indicated, SB-505124

(Tocris) at 10 or 50 µM, LDN-193189 (a gift from Paul Yu) at 1 or 0.5 µM, DMH1 (Selleck Chemicals, Germany) at 1 µM, cyclohexamide (Sigma) at 20 µg/ml and actinomycin D (Sigma) at 1 µg/ml. For TGF-β blocking experiments, the pan-TGF-β blocking antibody (1D11) and the control antibody (13C4) were used at 30 µg/ml (*Nam et al., 2008*).

## CRISPR/Cas9 knockout of SMAD1/5 and ACVR1/BMPR1A in NMuMG cells

From the wild-type NMuMG cells, a parental clone was selected that expressed robust junctional markers (TJP1 and CDH1) and underwent an efficient EMT in response to TGF-β. Two guide RNAs (see Key Resources Table) targeting the MH1 domain (SMAD1) and MH2 domain (SMAD5) were expressed from the plasmid pSpCas9(BB)−2A-GFP (PX458) (*Ran et al., 2013*) and used to knock out SMAD1 and SMAD5. NMuMG parental clone cells were simultaneously transfected with both plasmids, sorted for GFP expression, plated as single cells in 96-well plates and screened by sequencing to verify mutations in SMAD1 and SMAD5. Two knockout clones, ΔSMAD1/5 clone 1 and 2, were used in these studies. The same parental clone of NMuMG cells was also used to generate a line knocked out for ACVR1 and BMPR1A. The strategy was as described for the SMAD1/5 knockout and the guides are given in the Key Resources Table.

## Generation of cell lines stably expressing ACVR1-IPF

The InversePericam FKBP1A (IPF) fusion protein was amplified by PCR from the pCS2+zALK3 IPF (*Michel et al., 2011*) and cloned in-frame downstream of the human ACVR1 cDNA sequence in the pcDNA3.1 Hygro (+) vector (Thermo Fisher). MDCKII and NIH-3T3 cells were transfected with the ACVR1-IPF construct and selected with 400 µg/ml hygromycin or 40 µg/ml hygromycin, respectively. After selection, cells were FACS sorted for GFP expression. MDCKII ACVR1-IPF cells were maintained as a pool, while a single clone was isolated for NIH-3T3 cells. To test the functionality of ACVR1-IPF, NMuMG cells knocked out for ACVR1 and BMPR1A were transfected with empty pcDNA3.1 Hygro (+), ACVR1-IPF or FLAG-ACVR1 (*Daly et al., 2008*) as a positive control, and activity was monitored by their ability to induce phosphorylation of SMAD1/5.

## Generation and LED light photoactivation of Opto-receptors

The general design of the Opto receptors was as previously described (*Sako et al., 2016*). Opto-TGFBR1* and Opto-ACVR1 were generated by overlapping PCR (*Horton et al., 1990*) to include a N-terminal myristyolation domain, the intracellular domain of either human TGFBR1 (residues 149–503) or human ACVR1 (residues 147–509), a light-oxygen-voltage (LOV) domain from *Vaucheria frigida* (*Takahashi et al., 2007*) and a C-terminal HA-tag and cloned into the pCS2 expression plasmid (see *Supplementary file 1* and *2*). In the case of TGFBR1, the T204D point mutation was introduced that renders the kinase constitutively active (*Wieser et al., 1995*), thus generating the construct Opto-TGFBR1*. A kinase dead version of Opto-TGFBR1 was also generated in which K232 was mutated to R (*Wrana et al., 1994*). Furthermore, the GS-domain of ACVR1 ($^{189}$TSGSGS$^{194}$) was mutated to VAGAGA to generate Opto-ACVR1 GS-mut. NIH-3T3 cells were transfected with a total of 2 µg of plasmid DNA that included either 5 ng of GFP-SMAD3 (*Nicolás et al., 2004*) or 25 ng of Flag-SMAD1 (*Lechleider et al., 2001*) alone or in combination with 25 ng of Opto-TGFBR1* and/or 50 ng of Opto-ACVR1 (WT or GS-mut). We co-transfected the SMADs with the Opto-receptors to increase the range of the assay. Twenty-four hours post-transfection, cells were starved overnight in DMEM with 0.5% FCS. Cells were then left untreated or pre-treated with 0.5 µM LDN-193189 or 50 µM SB-505124 and then exposed to blue light from an LED array for 1 hr at 37°C in a humidified incubator. Control cells (i.e. in the dark) were wrapped in aluminium foil and placed in the same incubator.

## siRNAs and transfections

All siRNAs were purchased from Dharmacon/GE Health Care Life Sciences (UK) and are listed in *Supplementary file 3*. MDA-MB-231 and NMuMG cells were transfected with siRNAs at a final concentration of 20 nM with Interferin (Polyplus, France). Twenty four hours post-transfection, cells were starved overnight, and the following day cells were treated with TGF-β or BMP4 for 1 hr and RNA

and/or protein extracted. NMuMG cells were also treated with TGF-β for a further 24–48 hr to assess the effects of target gene knockdown on EMT.

## EMT assay

NMuMG or EpRas cells were plated on glass coverslips in six-well plates (200,000 or 75,000 cells, respectively). For NMuMG cells treated with small molecule inhibitors, the media was changed the day after plating to OptiMEM with 10 µg/ml insulin and the cells treated with 2 ng/ml TGF-β alone or in combination with 0.125 µM SB-431542, 1 µM LDN-193189 or DMH1 for the durations indicated. For knockdown experiments, NMuMG cells were transfected the day after plating with the indicated siRNAs. Twenty-four hours after transfection, the media was changed to OptiMEM with 10 µg/ml insulin and the following day, cells were treated with TGF-β for the durations indicated. For EpRas, cells were treated with 2 ng/ml TGF-β alone or in combination with 0.125 µM SB-431542 and 1 µM LDN-193189 the day after plating. EpRas cells were then split and re-plated at the initial splitting density in the presence of 2 ng/ml TGF-β alone or in combination with SB-431542 and LDN-193189 every 3 days.

## Antibodies, immunoblotting, immunoprecipitations and indirect immunofluorescence

All primary and secondary antibodies used are listed in the Key Resources Table. Western blots using whole cell extracts and immunoprecipitations followed by western blotting were as previously described (*Germain et al., 2000*; *Daly et al., 2008*). Indirect immunofluorescence of the ACVR1-IPF was performed after fixing cells in 4% formaldehyde for 5 min. Indirect immunofluorescence for CDH1 and TJP1 was performed after fixation in methanol:acetone (1:1) as previously described (*Nicolás and Hill, 2003*). Nuclei were counter stained with DAPI (0.1 µg/ml). Imaging was performed on a Zeiss Upright 780 confocal microscope. Z-stacks were acquired for all channels and maximum intensity projection images are shown.

## Live cell imaging

Live cell imaging was performed for MDCKII ACVR1-IPF and NIH-3T3 ACVR1-IPF cells on a Zeiss Invert 780 confocal microscope. Cells were plated on 35 mm MatTek dishes (MatTek, Ashland, MA, USA) and starved overnight in phenol-free, HEPES-buffered DMEM with 0.5% FCS. During imaging, the temperature was maintained at 37°C. Data were acquired every 15 min over a time course. At each time point, a z-stack was acquired, and maximum intensity z-projections were quantified with ImageJ.

## Flow cytometry

MDCKII ACVR1-IPF and NIH-3T3 ACVR1-IPF cells were treated with ligand ± inhibitors. Twenty four hours post treatment, cells were trypsinized, washed and analyzed for GFP/YPF fluorescence on a LSRII flow cytometer (BD Biosciences, San Jose, CA, USA), gated for viable, single cells. Treatment with FK506 (Sigma) was performed for 4 hr prior to analysis.

## Recombinant proteins, in vitro kinase assays and mapping of phospho-sites

Recombinant SMAD proteins were expressed in *E. coli* and purified as previously described (*Ross et al., 2006*). Recombinant intracellular domains of ACVR1, BMPR1A and TGFBR1 which were expressed in insect cells were purchased from Carna Biosciences Inc (Japan; see Key Resources Table). Radioactive kinase reactions were performed with varying amounts of receptor (25–200 ng) at 37°C for 1 hr in a 20 µl reaction volume with 50 mM Tris-Cl (pH 7.5), 50 mM NaCl, 5 mM MnCl$_2$ (ACVR1 and TGFBR1) or MgCl$_2$ (BMPR1A), 16.5 nM $^{32}$P-γ-ATP (Perkin Elmer, UK; NEG502A500UC) and either 200 µM or 50 µM cold ATP. Substrates were either the receptors themselves (autophosphorylation) or 2 µg of recombinant SMAD proteins. Reactions were stopped by adding Laemmli sample buffer and heating to 95°C for 5 min. Proteins were resolved on a NuPAGE Novex 4–12% Bis-Tris gradient gel (Thermo Fisher) and stained with Colloidal Blue (Thermo Fisher). Gels were destained, dried and radioactivity measured by autoradiography.

To map phosphorylated residues on SMAD1, radioactive kinase reactions were performed in triplicate with 200 ng ACVRI, 2 µg recombinant SMAD1, 200 µM cold ATP, 0.33 µM $^{32}$P-γ-ATP. For phospho-residue mapping, $^{32}$P-labeled SMAD1 was digested with trypsin, the peptides were resolved by HPLC with an acetonitrile gradient and the $^{32}$P-labeled peptides eluted. Edman sequencing and mass-spectrometry (Orbitrap Classic, Thermo Fisher) were then used to confirm phospho-residues, as described previously (*Campbell and Morrice, 2002*), with the addition of multi-stage activation during the MS2 analysis.

## Chromatin immunoprecipitations, ChIP-Seq and motif enrichment

Four million MDA-MB-231 or BT-549 cells were plated; 24 hr later, cells were starved overnight and the following day treated with TGF-β or BMP4. One 15 cm plate was used per immunoprecipitation. Chromatin immunoprecipitations, ChIP-seq library preparation, next generation sequencing and data analysis were performed in biological duplicate essentially as previously described (*Coda et al., 2017*). In brief, ChIP-seq was performed on an Illumina HiSeq2500 (Illumina, San Diego, CA, USA) generating 50 bp single end reads. Reads were aligned to the human GRCh37/hg19 genome assembly using BWA version 0.6 (*Li and Durbin, 2009*) with a maximum mismatch of 2 bases. Picard tools version 1.81 (http://sourceforge.net/projects/picard/) was used to sort, mark duplicates and index the resulting alignment bam files. Normalized tdf files for visualization purposes were created using IGVtools software (*Robinson et al., 2011*) (http://software.broadinstitute.org/software/igv/) by extending reads by 50 bp and normalizing to 10 million mapped reads per sample. Peaks were called by comparing stimulated samples to the respective untreated samples using MACS version 1.4.2 (*Zhang et al., 2008*), using mfold change parameters of between 5 and 30. Peaks called by MACS were annotated using the annotatepeaks command in the Homer software (*Heinz et al., 2010*) (http://homer.salk.edu/homer/).

Peaks with less than 20 tags in the pSMAD1/5 IP after TGF-β treatment or less than 30 tags in the SMAD3 IP after TGF-β treatment were excluded from the analysis. In addition, peaks that had less than 1 tag per 10 bp in either of the above conditions were also excluded. Finally a ratio was taken between the number of tags in the pSMAD1/5 IP and the number of tags in the SMAD3 IP after TGF-β treatment to determine the top 100 peaks with preferential SMAD1/5 binding. Of these, the top 50 peaks with the highest density of tags per 10 bases in the pSMAD1/5 IP after TGF-β treatment were used for more refined motif enrichment analysis and gene annotation.

Motif enrichment was performed using MEME (http://meme-suite.org/) with default parameters (zero or one occurrence per sequence, motifs between 6 and 50 bases in width).

## RNA-sequencing analysis in the NMuMG parental clone and ΔSMAD1/5 clone 1

NMuMG parental and ΔSMAD1/5 clone 1 were plated, starved the next day in OptiMEM with 10 µg/ml insulin and treated for a further 48 hr with 2 ng/ml TGF-β. Total RNA was extracted as previously described (*Grönroos et al., 2012*), DNase I (Qiagen, Germany) treated and cleaned up with RNeasy columns (Qiagen). Biological triplicate libraries were prepared using the TruSeq RNA Library Prep Kit (Illumina) and were single-end sequenced on an Illumina HiSeq 2500 platform. Sequencing yield was typically ~80 million strand-specific reads per sample. The RSEM package (version 1.2.31) (*Li and Dewey, 2011*) in conjunction with the STAR alignment algorithm (version 2.5.2a) (*Dobin et al., 2013*) was used for the mapping and subsequent gene-level counting of the sequenced reads with respect to Ensembl mouse GRCm.38.86 version genes. Normalization of raw count data and differential expression analysis was performed with the DESeq2 package (version 1.10.1) (*Love et al., 2014*) within the R programming environment (version 3.2.3) (*R Development Core Team, 2009*). Genes were first identified as differentially expressed in the parental clone if they had more than 10 reads in either the untreated or TGF-β treated samples and a fold change between untreated and TGF-β induced of > 1.5 or < 0.75 and FDR < 0.05. An interaction contrast was then used to determine differentially regulated genes after TGF-β treatment in the parental clone versus ΔSMAD1/5 clone 1. The resulting gene lists ranked by the Wald statistic were used to look for pathway and biological process enrichment using the Broad's GSEA Tool (*Subramanian et al., 2005*). Genes with a fold difference between the two clones after TGF-β treatment of > 1.5 or < 0.75 and an FDR < 0.05 were judged to be dependent on SMAD1/5.

## Public availability of data

The ChIP-seq data have been submitted to the NCBI Gene Expression Omnibus (GEO) under the accession number GSE92443. The RNA-seq data has been submitted to GEO under the accession number GSE103372.

## qPCR

Oligonucleotides used are listed in *Supplementary file 3*. Total RNA extraction and reverse transcription were performed as previously described (*Grönroos et al., 2012*). The cDNA was diluted 10-fold and then used for quantitative PCR (qPCR). All qPCRs were performed with Express Sybr Greener (Thermo Fisher) with 300 nM of each primer and 2 µl of diluted cDNA or eluted immunoprecipitated chromatin. Fluorescence acquisition was performed on a 7500 FAST machine (Thermo Fisher). Quantification for relative gene expression was done using the comparative Ct method with target gene expression normalized to *GAPDH*. Quantification for ChIPs was performed using a standard curve and presented normalized to input.

## Statistical analysis

Western blots, immunofluorescence experiments and ChIP-PCRs are representative of at least two biological replicate experiments. All qPCRs are the mean and SEM of three independent biological experiments except gene expression after actinomycin D treatment and stimulation with TGF-β3$^{WW}$ and TGF-β3$^{WD}$ and validation of RNA-sequencing results that are a representative of two independent experiments. Statistical analyses were performed with the unpaired Students T-Test, *p<0.05, **p<0.01, ***p<0.001, ns, non significant.

# Acknowledgements

We thank Lalage Wakefield for providing the TGF-β neutralizing antibody and the isotype-matched control, Paul Yu for LDN-193189, Christian Bökel for the pCS2+zALK3 IPF expression plasmid and Bob Lechleider for the FLAG-SMAD1 expression plasmid. We thank Nik Matthews, Greg Elgar and the Advanced Sequencing Facility for the next generation sequencing. We are grateful to the Francis Crick Institute Light Microscopy and Flow Cytometry facilities and to the Genomics Equipment Park. We thank Alex Bullock for very fruitful discussions and all the members of the Hill lab for useful comments on the manuscript. This work was supported by the Francis Crick Institute which receives its core funding from Cancer Research UK (FC001095), the UK Medical Research Council (FC001095), and the Wellcome Trust (FC001095). The development and characterization of TGF-β3$^{WD}$ in the Hinck laboratory was enabled by support provided by the NIH (GM58670 and CA172886).

# Additional information

## Funding

| Funder | Grant reference number | Author |
| --- | --- | --- |
| Francis Crick Institute | FC001095 | Anassuya Ramachandran<br>Pedro Vizan<br>Debipriya Das<br>Probir Chakravarty<br>Caroline S Hill |
| NIH Office of the Director | GM58670 | Andrew P Hinck |
| NIH Office of the Director | CA172886 | Andrew P Hinck |

The funders had no role in study design, data collection and interpretation, or the decision to submit the work for publication.

## Author contributions

Anassuya Ramachandran, Conceptualization, Data curation, Formal analysis, Investigation, Methodology, Writing—original draft, Writing—review and editing; Pedro Vizán, Conceptualization, Investigation; Debipriya Das, Investigation; Probir Chakravarty, Data curation, Formal analysis; Janis Vogt,

Methodology; Katherine W Rogers, Resources; Patrick Müller, Resources, Supervision, Writing—review and editing; Andrew P Hinck, Resources, Writing—review and editing; Gopal P Sapkota, Formal analysis, Supervision, Methodology, Writing—review and editing; Caroline S Hill, Conceptualization, Supervision, Funding acquisition, Writing—original draft, Project administration, Writing—review and editing

**Author ORCIDs**
Anassuya Ramachandran (iD) http://orcid.org/0000-0002-0123-6539
Katherine W Rogers (iD) http://orcid.org/0000-0001-5700-2662
Patrick Müller (iD) http://orcid.org/0000-0002-0702-6209
Gopal P Sapkota (iD) http://orcid.org/0000-0001-9931-3338
Caroline S Hill (iD) http://orcid.org/0000-0002-8632-0480

**Decision letter and Author response**
Decision letter https://doi.org/10.7554/eLife.31756.053
Author response https://doi.org/10.7554/eLife.31756.054

## Additional files

**Supplementary files**
• Supplementary file 1. Sequence of Opto-TGFBR1*.
DOI: https://doi.org/10.7554/eLife.31756.043

• Supplementary file 2. Sequence of Opto-ACVR1.
DOI: https://doi.org/10.7554/eLife.31756.044

• Supplementary file 3. List of oligonucleotides and siRNAs.
DOI: https://doi.org/10.7554/eLife.31756.045

• Supplementary file 4. Key resources table.
DOI: https://doi.org/10.7554/eLife.31756.046

• Transparent reporting form
DOI: https://doi.org/10.7554/eLife.31756.047

**Major datasets**
The following datasets were generated:

| Author(s) | Year | Dataset title | Dataset URL | Database, license, and accessibility information |
|---|---|---|---|---|
| Anassuya Ramachandran, Pedro Vizan, Debipriya Das, Probir Chakravarty, Janis Vogt, Andrew P Hinck, Gopal P Sapkota, Caroline S Hill | 2016 | TGF-$\beta$ signalling through SMAD1/5 is required for epithelial-to-mesenchymal transition | http://www.ncbi.nlm.nih.gov/geo/query/acc.cgi?acc=GSE92443 | Publicly available at the NCBI Gene Expression Omnibus (accession no: GSE92443). |
| Anassuya Ramachandran, Pedro Vizan, Debipriya Das, Probir Chakravarty, Janis Vogt, Katherine W Rogers, Patrick Müller, Andrew P Hinck, Gopal P Sapkota, Caroline S Hill | 2017 | TGF-$\beta$ utilizes a novel receptor activation mechanism to phosphorylate SMAD1/5 and regulate epithelial-to-mesenchymal transition | https://www.ncbi.nlm.nih.gov/geo/query/acc.cgi?acc=GSE103372 | Publicly available at the NCBI Gene Expression Omnibus (accession no: GSE103372). |

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
