## [Decision Letter]

Thank you for submitting your article "TGF-β uses a novel mode of receptor activation to phosphorylate SMAD1/5 and induce epithelial-to-mesenchymal transition" for consideration by *eLife*. Your article has been reviewed by three peer reviewers, one of whom is a member of our Board of Reviewing Editors and the evaluation has been overseen by Philip Cole as the Senior Editor. The reviewers have opted to remain anonymous.

The reviewers have discussed the reviews with one another and the Reviewing Editor has drafted this decision to help you prepare a revised submission.

Summary:

This is a compelling study that demonstrates a role for ACVR1 in the activation of Smad1/5 by TGFbeta. The mechanism is reported to be mediated by TGFBR1 mediated phosphorylation and activation of ACVR1. The authors show that the Smad1/5 arm of the TGF β signaling pathway leads to the regulation of a fraction of the total TGF β responsive genes. Moreover, the authors associate this gene expression program with EMT. In conclusion, this is an excellent, well written manuscript that provides compelling evidence to explain a known paradox in TGFB signaling, the transient induction of pSMAD1/5, and shows that it is biologically important.

Essential revisions:

1) While 10 µM SB inhibits both pSmad1/5 and pSmad2, and 0.25 μm still inhibits both but possibly inhibiting more pSmad1/5 than pSmad2 (no quantification done), it is too strong to conclude that TGFBR1 is differentially required for SMAD1/5 versus SMAD2 phosphorylation. More evidence is required before such a conclusion can be made.

2) Figure 2. Since TGFBR1 opto-activation alone can robustly induce phosphorylation of SMAD1, it cannot be concluded that TFGBR1 + ACVR1 opto-activation of pSMAD1 is by TGFBR1 activating ACVR1. This conclusion in the Discussion section, and in the Abstract (possibly elsewhere) should not be made based on these results.

3) Is ACVR1-IPF functional? The cells used already induce pSMAD1 in response to TGFB treatment – the endogenous receptors are therefore functional.

4) The authors speculate in the Discussion that the prolonged fluorescence of ACVR1-IPF reflects continued activation of ACRV1 in the absence of pSMAD1 and that a phosphatase likely dephosphorylates pSMAD1 in this system. Do the authors know that the ACVR1-IPF conformation change of cpYFP is reversible in their system? Perhaps ACVR1 can no longer be activated, but cpYFP fluorescence persists?

5) The optogenetic studies (Figure 2 and Figure 4) require a control using a TGFBR1 kinase dead mutant to show that TGFBR1 kinase activity is required for the activation of ACVR1. This is necessary to exclude the possibility that ACVR1 activation is mediated by conformational interactions alone.

6) Figure 2 and subsection “ACVR1 is activated by TGFBR1 in vitroand in vivo*”*: Light-inducible FLAG-Smad1 phosphorylation was reduced when the GS domain mutant of Opto-ACVR1 was used instead of Opto-ACVR1, but light-inducible endogenous Smad1 phosphorylation was not (Figure 2, four lanes at the right side). In addition, co-expression of Opto-ACVR1 with Opto-TGFR1* failed to affect endogenous Smad1 phosphorylation. These results do not support the authors' conclusion. Why was co-transfected FLAG-Smad1 used in these studies since activation of the BMP type I receptor can be monitored by endogenous Smad1.

7) It has been shown that TGFB-WD cannot bind TGFBR1 or R2, but can it bind ACVR1? Has this been tested? If it has not been tested, then it is possible that TBR1 and ACVR1 are in the same complex.

8) Conflicts with an earlier study: Daly et al., (2008). Figure 5, subsection “Mapping the binding sites on chromatin for TGF-b-induced pSMAD1/5 reveals that *ID* genes are major transcriptional targets of this pathway” and subsection “Combinatorial signaling downstream of TGF-β”: The authors previously reported that TGF-b-induced Smad1/5 phosphorylation does not lead to transcriptional activation via BMP-responsive element, because Smad2/3-Smad1/5 mixed complexes are formed. In addition, they stated that Smad1 coimmunoprecipiated with Smad4 only after stimulation with BMP-4 but not TGF-b in EpH4 cells (Daly et al., 2008). Now they present data in which TGF-b induces formation of a Smad1/5-Smad4 complex. This is also confusing to readers. They should mention and discuss their previous results in the main text.

Figure 6: The authors previously reported that knockdown of Smad1/5 did not affect TGF-b-induced EMT in EpRas cells (based on cell morphology and mesenchymal marker expression) (Daly et al., 2008). Now they state that knockout of Smad1/5 inhibited TGF-b-induced EMT in NMuMG cells (based on localization of ZO-1 and E-cadherin). This is quite confusing to readers. Localization of ZO-1 and E-cadherin was partially disrupted and cell morphologies were changed by TGF-b in Smad1/5 knockout cells (Figure 6, compare with Figure 7). The present statement is thus misleading. Only a subset of the EMT processes appears to be affected by loss of Smad1/5. At least, expression of epithelial as well as mesenchymal markers should be examined.

---

## [Author Response]

Essential revisions:1) While 10 µM SB inhibits both pSmad1/5 and pSmad2, and 0.25 μm still inhibits both but possibly inhibiting more pSmad1/5 than pSmad2 (no quantification done), it is too strong to conclude that TGFBR1 is differentially required for SMAD1/5 versus SMAD2 phosphorylation. More evidence is required before such a conclusion can be made.

The reviewer is correct that we cannot conclude that TGFBR1 is differentially required for SMAD1/5 versus SMAD2 phosphorylation based solely on the observation that the low dose of SB-431542 inhibits phosphorylation of SMAD1/5 more effectively than SMAD2 phosphorylation. We have now removed this sentence (see subsection “TGF-β-induced SMAD1/5 phosphorylation requires the kinase activity of two different type I receptors”).

The main aim of using the drugs was to establish conditions where we could inhibit TGF-β-induced phosphorylation of SMAD1/5, whilst having a minimal effect on SMAD2 phosphorylation. We show this in three different cells lines (in MDA-MB-231s (Figure 1); in NMuMGs (Figure 6—figure supplement 3) and in EpRas cells (Figure 6) using a combination of LDN-193189 and low dose SB-431542 (either 0.25 µM or 0.125 µM depending on the cell line).

The reviewers are referring specifically to the result in the MDA-MB-231 cells (Figure 1), where it is evident that the 0.25 µM SB-431542 alone did have a small inhibitory effect on SMAD2 phosphorylation, although a stronger effect on SMAD1/5 phosphorylation. Because the quality of the original pSMAD2 blot was not very high, we have now re-done this experiment and replaced the panel (see new Figure 1). In the new Figure it is clear that the 0.25 µM SB-431542 dose does have a small effect on pSMAD2 levels, but a stronger effect on pSMAD1/5 levels. In combination with LDN-193189 it is sufficient to completely inhibit pSMAD1/5, whilst having no additional effect on pSMAD2.

Given that we show the result in three different cell lines, we consider that the result in robust and the conclusion is valid.

2) Figure 2. Since TGFBR1 opto-activation alone can robustly induce phosphorylation of SMAD1, it cannot be concluded that TFGBR1 + ACVR1 opto-activation of pSMAD1 is by TGFBR1 activating ACVR1. This conclusion in the Discussion section, and in the Abstract (possibly elsewhere) should not be made based on these results.

The ability of activated Opto-TGFBR1* to induce phosphorylation of SMAD1 in response to light was unexpected. However, because we were able to show that this SMAD1 phosphorylation was inhibited by both SB-505124 (which inhibits TGFBR1) and LDN-193189 (which only inhibits the BMP type I receptors), we were confident concluding that this resulted from the light induced TGFBR1* activating endogenous ACVR1 (and possibly also BMPR1A/B), which then phosphorylate SMAD1. Importantly, when we combined Opto-TGFBR1* with Opto-ACVR1 (which has no activity on its own) we saw a robust increase in SMAD1 phosphorylation, as the exogenous FLAG-SMAD1, that we added to increase the range of the assay, was strongly phosphorylated.

We have now optimized the assay conditions further and have realised that the amount of SMAD1 phosphorylation that we see with Opto-TGFBR1* alone is dependent on the amount of this receptor transfected and also on the transfection efficiency of the experiment. We have now repeated the experiments and present the new data in Figure 2 and E. In these conditions there is minimal phosphorylation of SMAD1 when Opto-TGFBR1* is used alone, but robust phosphorylation when used together with Opto-ACVR1. Taking these results together with the IPF results (Figure 4) and the in vitro phosphorylation results (Figure 2), we are confident that we can conclude that TGFBR1 activates ACVR1 that phosphorylates SMAD1/5.

3) Is ACVR1-IPF functional? The cells used already induce pSMAD1 in response to TGFB treatment – the endogenous receptors are therefore functional.

This is a very valid point. We know that it is activated in response to ligand, and we can see it being internalised into vesicles (evident in the Videos), but we had not demonstrated that it can phosphorylate SMAD1/5 downstream.

To do this cleanly we needed a cell line null for ACVR1 and BMPR1A. We have now generated such a line in NMuMGs using CRISPR/Cas9. Transfection of the ACVR1-IPF results in phosphorylation of SMAD1/5 (see new Figure 4—figure supplement 1 and subsection “TGF-β induces ACVR1 activation in vivoin a TGFBR1-dependent manner”). Thus, we can conclude that the ACVR1-IPF is active.

4) The authors speculate in the Discussion that the prolonged fluorescence of ACVR1-IPF reflects continued activation of ACRV1 in the absence of pSMAD1 and that a phosphatase likely dephosphorylates pSMAD1 in this system. Do the authors know that the ACVR1-IPF conformation change of cpYFP is reversible in their system? Perhaps ACVR1 can no longer be activated, but cpYFP fluorescence persists?

The reviewers raise a very valid point. We do not think that the conformation change of cpYFP is reversible in our system. As we observe that the levels of fluorescence continue to accumulate over time when the cells are stimulated with ligand and does not plateau, we conclude that new surface receptor is continuously being activated during the course of the experiment.

5) The optogenetic studies (Figure 2 and Figure 4) require a control using a TGFBR1 kinase dead mutant to show that TGFBR1 kinase activity is required for the activation of ACVR1. This is necessary to exclude the possibility that ACVR1 activation is mediated by conformational interactions alone.

We have now done this control and show the results in new Figure 2—figure supplement 1. The results show that the kinase-dead Opto-TGFBR1 has no activity in this assay at all. This result complements the experiment we had already done showing that we can inhibit the light-induced phosphorylation of SMAD1/5 by Opto-TGFBR1* and Opto-ACVR1 with the TGFBR1 inhibitor SB-505124 (see new Figure 2).

*6) Figure 2 and subsection “ACVR1 is activated by TGFBR1* in vitro *and* in vivo*”: Light-inducible FLAG-Smad1 phosphorylation was reduced when the GS domain mutant of Opto-ACVR1 was used instead of Opto-ACVR1, but light-inducible endogenous Smad1 phosphorylation was not (Figure 2, four lanes at the right side). In addition, co-expression of Opto-ACVR1 with Opto-TGFR1* failed to affect endogenous Smad1 phosphorylation. These results do not support the authors' conclusion. Why was co-transfected FLAG-Smad1 used in these studies since activation of the BMP type I receptor can be monitored by endogenous Smad1.*

We used FLAG-SMAD1 in the assays because, in our experience of overexpression of wild type receptors in many different cell lines, we had previously found that they rarely phosphorylated endogenous SMADs very well and were much better assayed by co-transfecting the substrate SMAD, possibly because of relatively low transfection efficiency. (For a nice example of this see Figure 7 in Randall et al., (2004)). However, initial experiments indicated that we could see some phosphorylation of endogenous SMAD1/5 by the Opto receptors, but we retained the FLAG-SMAD1 in the assay as it substantially extends the range of the assay. This is now mentioned in the text (see subsection “ACVR1 is activated by TGFBR1 in vitroand in vivo” and subsection “Generation and LED light photoactivation of Opto-receptors”).

As explained under point 2, we have now found conditions where the Opto-TGFBR1* alone has little effect on SMAD1/5 phosphorylation, but when combined with Opto-ACVR1 can induce robust phosphorylation upon light induction. This is predominantly seen as phosphorylation of the FLAG-SMAD1 and the result is very clear (new Figure 2).

We have not redone the GS domain experiment as we consider that the result was clear, in that we can show that the phosphorylation of FLAG-SMAD1 that results from Opto-ACVR1 in the presence of Opto-TGFBR1* is dependent on the presence of an intact GS domain. Note that the Opto-TGFBR1*-induced phosphorylation of endogenous SMAD1 by light in this experiment will obviously not be inhibited when we use the Opto-ACVR1 GS domain mutant, as it is mediated by endogenous BMP type I receptors (see explanation under point 2).

7) It has been shown that TGFB-WD cannot bind TGFBR1 or R2, but can it bind ACVR1? Has this been tested? If it has not been tested, then it is possible that TBR1 and ACVR1 are in the same complex.

Just to clarify, the mutations in the TGF-β3 D subunit render it unable to bind TGFBR1 or TGFBR2. However, the TGF-β3 WD heterodimer obviously can bind these receptors, as it does so through the wild type subunit (TGF-β3 W) binding a type I/type II dimer. This is the essence of Huang et al., 2011 paper that we refer to.

With reference to the possibility that TGF-β3 WD might bind ACVR1. Studies from the Hinck lab a number of years ago showed that TGF-β3 WW cannot bind ACVR1. Given this, it is extremely unlikely that the mutated D subunit, which was designed to abolish receptor binding (to TGFBR1 or TGFBR2), would be able to bind ACVR1. This is now mentioned in the text (see subsection “TGF-β leads to clustering of ACVR1 and TGFBR1”).

8) Conflicts with an earlier study: Daly et al., (2008). Figure 5, subsection “Mapping the binding sites on chromatin for TGF-b-induced pSMAD1/5 reveals that ID genes are major transcriptional targets of this pathway” and subsection “Combinatorial signaling downstream of TGF-β”: The authors previously reported that TGF-b-induced Smad1/5 phosphorylation does not lead to transcriptional activation via BMP-responsive element, because Smad2/3-Smad1/5 mixed complexes are formed. In addition, they stated that Smad1 coimmunoprecipiated with Smad4 only after stimulation with BMP-4 but not TGF-b in EpH4 cells (Daly et al., 2008). Now they present data in which TGF-b induces formation of a Smad1/5-Smad4 complex. This is also confusing to readers. They should mention and discuss their previous results in the main text.

We are sorry for the confusion. The reviewer is correct. In our 2008 paper we were unable to detect any complexes between SMAD1/5 and SMAD4 in response to TGF-β in EpH4 cells, and we thought that this explained why the BMP responsive element (BRE) could not be activated in response to TGF-β. In a subsequent study (Gronroos et al., 2012) we could detect a low level of SMAD1/4–SMAD4 complexes by IP in MDA-MB-231 cells (see Figure 5 in that paper) and showed that the inability of TGF-β to activate the BRE was actually dependent on SMAD3 (Figure 7 in Gronroos et al.,). We concluded that a complex comprising pSMAD1/5 and pSMAD3 inhibits the activity of the BRE.

Upon further optimization of the IPs, we can now show clearly that TGF-β can induce pSMAD1/5–SMAD4 complexes as well as pSMAD2/3–SMAD4 and pSMAD2/3–pSMAD1/5 complexes in MDA-MB-231 cells (see Figure 5 in current paper).

We have amended the text to mention our previous work (see subsection “Mapping the binding sites on chromatin for TGF-β-induced pSMAD1/5 reveals that *ID* genes are major transcriptional targets of this pathway**”**).

Figure 6: The authors previously reported that knockdown of Smad1/5 did not affect TGF-b-induced EMT in EpRas cells (based on cell morphology and mesenchymal marker expression) (Daly et al., 2008). Now they state that knockout of Smad1/5 inhibited TGF-b-induced EMT in NMuMG cells (based on localization of ZO-1 and E-cadherin). This is quite confusing to readers. Localization of ZO-1 and E-cadherin was partially disrupted and cell morphologies were changed by TGF-b in Smad1/5 knockout cells (Figure 6, compare with Figure 7). The present statement is thus misleading. Only a subset of the EMT processes appears to be affected by loss of Smad1/5. At least, expression of epithelial as well as mesenchymal markers should be examined.

Again, we are sorry for the confusion. In our previous work in 2008 in EpRas cells, we got rather inefficient knockdown of SMAD1/5 by siRNAs and could not reproducibly show an effect on TGF-β-induced EMT. We reported this as data not shown. Now, with the use of the drug combinations in these EpRas cells (note that LDN-193189 was not available in 2008), we can clearly show that TGF-β-induced SMAD1/5 is required for a full EMT. We have also shown this in NMuMG cells using CRISPR to knock out SMAD1/5, siRNAs and also the drug combinations. We now mention this in the text (see subsection “TGF-β-induced SMAD1/5 signaling is required for EMT through induction of ID1”).

We have now stressed in the text that SMAD1/5 is required for a full EMT (see subsection “Combinatorial signaling downstream of TGF-β”). The reviewer is right in that ZO-1 and E-cadherin are partially disrupted upon TGF-β stimulation in the SMAD1/5 knockout cells and cell morphologies were changed.

We agree that it is important to show some mesenchymal genes as well as epithelial ones. In our RNA-seq of the wild type versus SMAD1/5 knockout NMuMG cells we observed a lower induction of *Acta2* (smooth muscle actin) and *Fn1* in the knockout cells relative to the wild type. These genes are highlighted now in red in Figure 6—source data 1, Sheet 2 and we have validated the result by qPCR (see new Figure 6—figure supplement 2). Also note that EMT came up as a relevant pathway when comparing the TGF-β-induced transcriptome of wild type and SMAD1/5 knockout cells (see Figure 6—source data 1, Sheet 3).